# A systematic review of *in vivo* stretching regimens on inflammation and its relevance to translational yoga research

**Dennis Muñoz-Vergara**[1,2]*, **Weronika Grabowska**[1], **Gloria Y. Yeh**[2,3], **Sat Bir Khalsa**[2,4], **Kristin L. Schreiber**[5], **Christene A. Huang**[6], **Ann Marie Zavacki**[7], **Peter M. Wayne**[1,2]

1 Division of Preventive Medicine, Brigham and Women's Hospital, Harvard Medical School, Boston, MA, United States of America, 2 Osher Center for Integrative Medicine, Brigham and Women's Hospital, Harvard Medical School, Boston, MA, United States of America, 3 Division of General Medicine and Primary Care, Beth Israel Deaconess Medical Center, Harvard Medical School, Boston, MA, United States of America, 4 Department of Medicine, Brigham and Women's Hospital, Harvard Medical School, Boston, MA, United States of America, 5 Department of Anesthesiology, Perioperative and Pain Medicine, Brigham and Women's Hospital, Harvard Medical School, Boston, MA, United States of America, 6 The Department of Surgery, School of Medicine, University of Colorado, Denver, CO, United States of America, 7 Division of Endocrinology, Diabetes, and Hypertension, Brigham and Women's Hospital, Harvard Medical School, Boston, MA, United States of America

* dmunozvergara@bwh.harvard.edu

**Data Availability Statement:** All relevant data used in this systematic review is included in the manuscript, tables, figures, and in supplementary/

## Abstract

### Objective

To conduct a systematic review evaluating the impact of stretching on inflammation and its resolution using *in vivo* rodent models. Findings are evaluated for their potential to inform the design of clinical yoga studies to assess the impact of yogic stretching on inflammation and health.

### Methods

Studies were identified using four databases. Eligible publications included English original peer-reviewed articles between 1900–May 2020. Studies included those investigating the effect of different stretching techniques administered to a whole rodent model and evaluating at least one inflammatory outcome. Studies stretching the musculoskeletal and integumentary systems were considered. Two reviewers removed duplicates, screened abstracts, conducted full-text reviews, and assessed methodological quality.

### Results

Of 766 studies identified, 25 were included for synthesis. Seven (28%) studies had a high risk of bias in 3 out of 10 criteria. Experimental stretching protocols resulted in a continuum of inflammatory responses with therapeutic and injurious effects, which varied with a combination of three stretching parameters—duration, frequency, and intensity. Relative to injurious stretching, therapeutic stretching featured longer-term stretching protocols. Evidence of pro- and mixed-inflammatory effects of stretching was found in 16 muscle studies. Evidence

supportive information. Files have been uploaded accordingly.

**Funding:** Financial support for the work was provided by grant T32 AT00051 from the National Center for Complementary and Integrative Health (NCCIH) to DMV, grant K24AT009282 from NCCIH to PMW, and grant R01DK044128 from the National Institute of Diabetes and Digestive and Kidney Diseases (NIDDK) to AMZ, grant K24AT009465 from NCCIH to GYY. The funders had no role in study design, data collection and analysis, decision to publish, or preparation of the manuscript. The specific roles of these authors are articulated in the 'author contributions' section.
T32AT00051— https://www.nccih.nih.gov
K24AT009282— https://www.nccih.nih.gov
K24AT009465— https://www.nccih.nih.gov
R01DK044128— https://www.niddk.nih.gov.

**Competing interests:** I have read the journal's policy and the authors of this manuscript have the following competing interests: Peter Wayne is the founder and sole owner of the Tree of Life Tai Chi Center. Peter Wayne's interests were reviewed and managed by the Brigham and Women's Hospital and Partner's HealthCare in accordance with their conflict of interest policies. This does not alter our adherence to PLOS ONE policies on sharing data and materials. No other authors have any potential conflicts to disclose.

of pro-, anti-, and mixed-inflammatory effects was found in nine longer-term stretching studies of the integumentary system.

## Conclusion

Despite the overall high quality of these summarized studies, evaluation of stretching protocols paralleling yogic stretching is limited. Both injurious and therapeutic stretching induce aspects of inflammatory responses that varied among the different stretching protocols. Inflammatory markers, such as cytokines, are potential outcomes to consider in clinical yoga studies. Future translational research evaluating therapeutic benefits should consider *in vitro* studies, active vs. passive stretching, shorter-term vs. longer-term interventions, systemic vs. local effects of stretching, animal models resembling human anatomy, control and estimation of non-specific stresses, development of *in vivo* self-stretching paradigms targeting myofascial tissues, and *in vivo* models accounting for gross musculoskeletal posture.

## Introduction

Stretching is an integral component of mind-body exercises such as yoga, tai chi, and qigong [1–3]. While clinical studies support the use of mind-body exercises for a range of health conditions, including chronic pain [4], metabolic [5], and affective disorders [6], few experimental studies have tried to isolate the impact of stretching from other potentially therapeutic components (e.g., focused attention, breathing, imagery, psychosocial support) [7], Multifactorial additive designs and/or dismantling studies represent possible experimental clinical approaches for isolating and mechanistically evaluating the therapeutic impact of a single component, such as stretching, from other intervention components. However, these approaches can be costly and premature, especially when preclinical data related to dose and key mediating physiological processes are still poorly understood [8].

One alternative preliminary strategy to inform future mechanistic clinical research design is to leverage *in vivo* rodent studies. A significant body of rodent research has explored the biological basis of both the therapeutic and injurious effects of stretching on several physiological systems. However, these studies have not been systematically reviewed to inform clinical research [9,10]. This systematic review draws on this body of scientific literature with a specific focus on *in vivo* rodent studies that have investigated the impact of stretching on pro- and anti-inflammatory processes, as well as processes related to inflammation resolution, in the musculoskeletal (MSK) and integumentary systems. We focus on inflammation because it is an evolutionarily preserved mechanism involved in tissue remodeling (e.g., tissue repair) [11–13], it is believed to play a central role across the spectrum of therapeutic and injurious stretching, and it has also been observed to change in response to physical activity and mind-body movement practices like yoga and tai chi [14].

More broadly, through this systematic review of findings from controlled preclinical studies, we aim to inform a basic framework for designing and interpreting clinical studies that evaluate the impact of yogic stretching on inflammation and health. Accordingly, we highlight five features of rodent stretching studies that are particularly relevant to translational research: 1) Use of passive vs. active stretching techniques; 2) Stretching parameters (intensity, duration, frequency, and whole-body posture); 3) Use of shorter- vs. longer-term stretching protocols; 4) Use of injurious vs. therapeutic stretching techniques; and 5) Relevance of inflammatory

outcomes, including macroscopic, histopathologic, genetic, immune cells/particles sorting, cytokines, and protective lipids (e.g., resolvins). Our discussion includes an exploration of a coordinated and bidirectional translation model of research, integrating pre-clinical and human studies to elucidate biological processes and guide clinical applications of multimodal interventions that include stretching, with an emphasis on yoga, in health and rehabilitation.

## Methods

### Protocol elaboration

The protocol for this systematic review follow the template of the Systematic Review Centre for Laboratory Animal Experimentation [15].

### Information sources and electronic search strategy

The studies included in this systematic review were identified using PubMed, Embase, the Web of Science search engines, and the Harvard University library search platform (Hollis). A broad search strategy was used to identify all relevant studies. PubMed and Embase included citations through April 22, 2020, and Web of Science through May 7, 2020. Hollis identified six additional studies. Keywords used to guide search terms across all three databases are listed in Table 1. A complete electronic search strategy listed in S1 Table.

### Study inclusion and exclusion criteria

Studies included investigated the effect of different stretching techniques (i.e., passive and/ or active stretching techniques with injurious and/or therapeutic intent, using either single or multiple bouts of stretching, and having a short- or long-term duration); administered to the MSK and integumentary systems of whole-rodent models (i.e., *in vivo*); and evaluating at least one outcome related to inflammation using at least one of the following five laboratory technique groups: microscopic, genetic, cell/particle sorting, enzymatic, and macroscopic techniques. Healthy populations of rodents (i.e., rats and mice) of any gender or age were considered. Inclusion also required either an independent non-stretched control group or the use of matched non-stretched contralateral muscles. Exclusion criteria included rodent studies stretching other body systems (e.g., reproductive or respiratory systems), *in vitro* and *ex vivo* paradigms, and clinical trials. In addition, eligible publications included all original peer-reviewed articles written in English between 1900 –May 2020. Pre-1900 articles were excluded because the laboratory techniques to evaluate inflammatory outcomes were not yet fully developed. Editorial materials, book and book chapters, biographical items, reviews, notes, letters, or social media news were excluded.

**Table 1. Summary keywords employed in the search strategy.**

| Animal | Stretching | Inflammation |
|--------|-----------|--------------|
| *In vivo* | Stretching exercise | Pro-resolving mediators |
| Rat | Passive stretching | Cytokine |
| Mouse | Active stretching | Muscle inflammation |
| | Stretch injury | Connective tissue inflammation |
| | Mechanical stretch | |

## Outcome measures

Eligible inflammation-related outcomes included a broad array of measures using: microscopic (e.g., histopathology), genetic (e.g., quantitative reverse transcription polymerase chain reaction—RT-qPCR), cell/particle-sorting (e.g., flow cytometry), enzymatic (e.g., ELISA), and macroscopic techniques (e.g., ultrasound).

## Duplicates removal, screening, and data extraction process

Two reviewers used the Covidence© software to remove duplicates, screen abstracts, and full-text reviews from articles identified (DMV and WG). A third reviewer resolved discrepancies (PW).

The data extraction template contained nine domains: authors, study aims, type and intention of stretching, animal model information (strain, sex, species, age, N), intervention, stretching parameters, inflammatory outcomes, experimental groups, and results. The study team agreed on these fields, and the data extraction was conducted by two study members (DMV and WG) and reviewed by two independent reviewers (PW and GY).

## Quality assessment

The SYRCLE's risk of bias tool for animal studies was used to assess the reported methodological quality of included studies [16]. Each criterion was classified as low, high, or unclear risk of bias.

# Results

## Search results

Our search strategy identified 766 studies. After removing duplicate citations, 695 publications were screened using titles and abstracts. Fifty-eight studies had a full assessment of eligibility. After excluding ineligible articles, 25 studies underwent a complete systematic synthesis (Fig 1).

## Quality assessment

Fig 2 summarized the quality assessment for the 25 articles included in the systematic review. Eight categories exhibited a low risk of bias in more than 50% of the studies. Outcome assessment blinding and other sources of bias were high and unclear risk of bias, respectively.

## Categorization of study features

Across the 25 studies, there was significant heterogeneity in the study designs used. We categorized studies according to key characteristics of reported data: passive versus active stretching, stretching parameters, stretching protocol (shorter- versus longer-term), injurious versus therapeutic intent, and rodent species. Table 2 elaborates on specific terms used in the following subsections.

**Passive versus active stretching.** Twenty-two studies used passive stretching, satisfying the definition of a movement applied by an external force [20–41]. All passive stretching studies anesthetized the animals to manual stretching or stretching achieved with a mechanical device [20–41]. Of the 22 studies, fourteen explored the effect of dynamic muscle activation with a protocol of lengthening and shortening contractions/cycles. One study directly stimulated the nerve tissue associated with muscles [24]. A third study performed passive elongation under anesthesia [20]. The remaining six studies examined the mechanical effect of stretching on the integumentary system of rats and mice. Of note, five studies used devices either on top

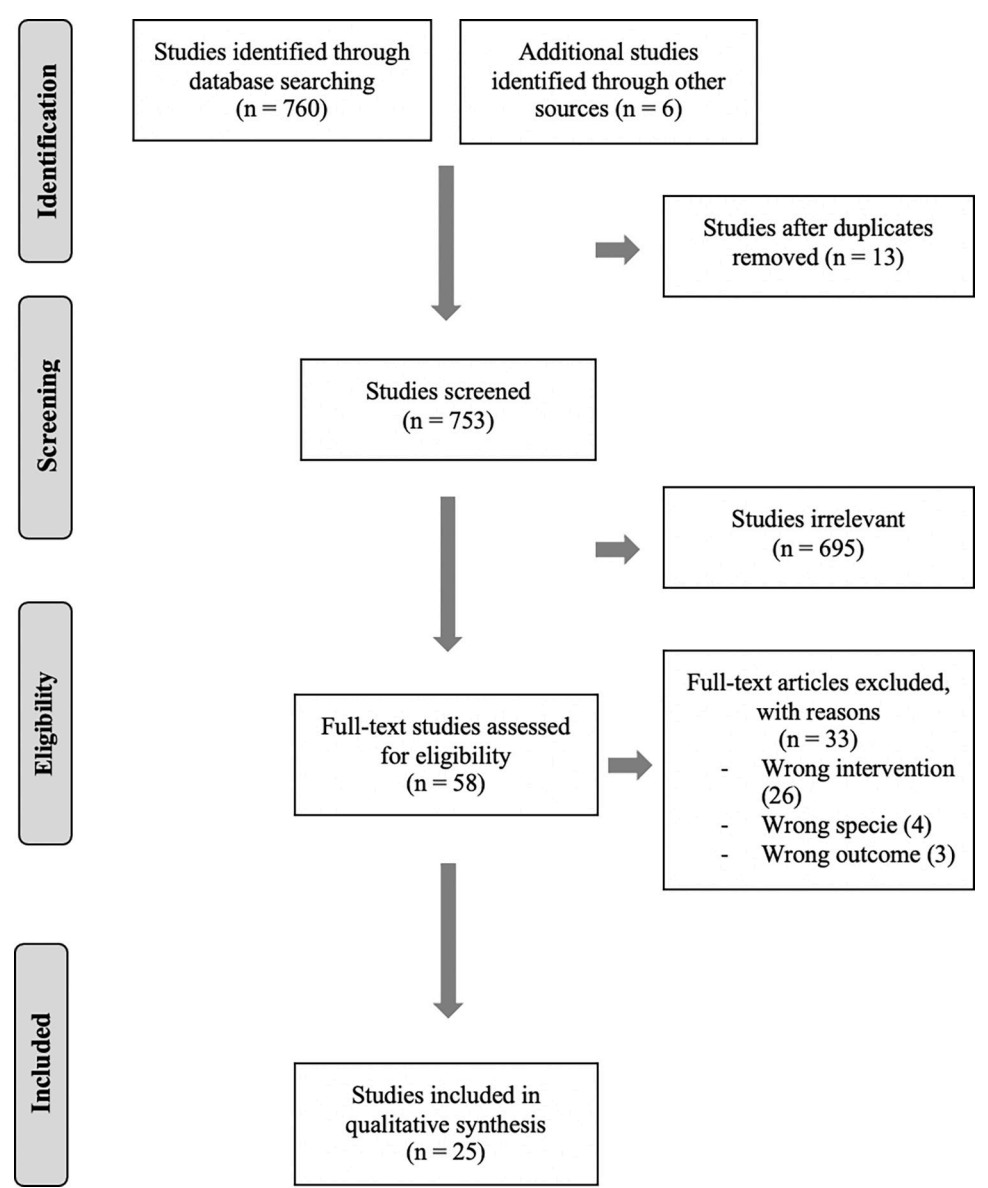

**Fig 1. PRISMA flow diagram.** Study screening flow chart for studies identified in the systematic literature review process.

of the skin or subcutaneously to deliver a static stretching protocol [25–28,40]. Four studies performed an active stretching intervention by lifting rats and mice by the tail until reaching an approximately 45˚ angle and increasing the distance between the shoulders and hips by ~25% [8,41–43].

   **Stretching parameters (intensity, duration, frequency, and posture).**   *Intensity* is defined as the magnitude of force generated during stretching. *Duration* corresponds to the amount of time in which the stretching occurs. *Frequency* refers to the number of stretching bouts in a given time, and *posture* is defined as the spatial body position during stretching [17]. Fourteen studies that stretched the musculoskeletal system controlled three out of four stretching parameters (intensity, duration, and frequency) [20,21,24,29–39]. One only controlled two of the four parameters (duration and frequency) [22], and another controlled for

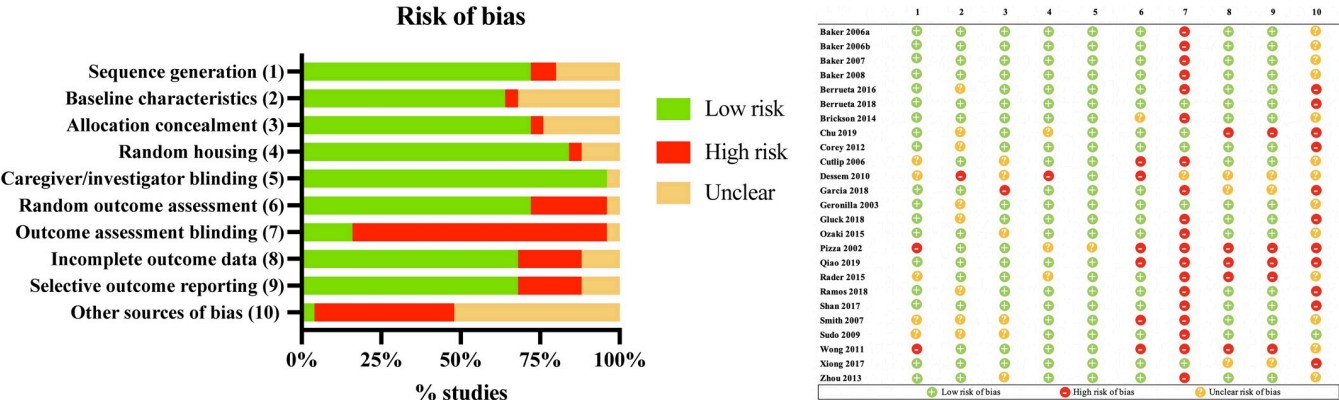

**Fig 2. Methodological quality of studies using SYRCLE's risk of bias assessment tool.** (A) The risk of sequence generation, baseline characteristics, and other biases was assessed for the studies included in this review. Eight categories have a low risk of bias in more than 50% of the studies, except of outcome assessment blinding and other sources of bias. (B) Studies fulfilling the criteria of: (1) Sequence generation; (2) Baseline characteristics; (3) Allocation concealment; (4) Random housing; (5) Caregiver/investigator blinding; (6) Random outcome assessment; (7) Outcome assessment blinding; (8) Incomplete outcome data; (9) Selective outcome reporting; (10) Other sources of bias.

one parameter (frequency) [23]. Three studies that stretched the skin and connective tissue held constant three out of four stretching parameters (intensity, duration, and frequency) [26–28]. In contrast, six studies controlled two out of four parameters (duration and frequency) [8,25,40–43] The posture parameter was not included by any of the studies included in this systematic review.

**Stretching protocol (shorter- versus longer-term*).*** We defined a shorter-term stretching intervention as one lasting between 2 seconds and 20 minutes with one or multiple bouts within 48 hours. Sixteen studies used a shorter-term stretching protocol. Of these, 15 studies performed the protocol on muscle, and one targeted the skin [20–24,29–33,35–39,41]. By comparison, we define a longer-term stretching intervention as one repeated on multiple occasions per week for 1–5 weeks. Ten studies used longer-term stretching protocols, with one targeting muscles and nine targeting skin [8,25–28,34,40–43].

**Table 2. Terminology and definition.**

| Term | Definition |
| --- | --- |
| Active stretching (AS) [17,18] | Refers to a movement applied by an external and/or internal force causing the interaction of the actin and myosin filaments of muscle due to the activation of muscle innervation and targeting the range of motion, i.e., flexibility. |
| Passive stretching (PS) [17] | Refers to a movement applied by an external force causing the elongation of the myofascial and integumentary systems beyond their resting length and targeting the range of motion, i.e., flexibility. |
| Stretch-shortening contractions/ cycles (SSCs) [19] | Refers to the muscle action when active muscle lengthening is immediately followed by active muscle shortening. This combination of eccentric and concentric contractions is one the most common type of muscle action during locomotion. |
| *In vivo* model of an acute muscle stretch injury [20–23] | These procedures include: A muscle stretch injury applied by a joint torque system which precisely controls the amount of tendon shortening; Muscle elongation without surgical exposure; Muscle tetanic contractions through electrical stimulation. |
| *In vivo* model of an acute nerve stretch injury [24] | Refers to the expose of nerves for stretching (e.g., median nerve between *flexor digitorum superficialis* and *flexor carpi radialis*). |
| Skin-stretching device model [25–28] | Refers to the use of devices attached to the skin or implanted under the skin to deliver controlled stretching to the skin and surrounding connective tissue. |

**Injurious versus therapeutic stretching.** Eighteen studies intended to use an injurious passive stretching model. Of these, thirteen applied it directly to muscle and two via muscle innervation [20–24,29–36,38,39] The remaining three studies targeted the skin and the connective tissue to examine the injurious effect of stretching with two different mechanical stretching devices delivering static stretching [25,27,40]. Seven studies investigated the therapeutic effect of stretching. One study employed a muscle model using one bout of lengthening contractions, followed by either passive stretches or isometric contractions [37]. Six studies used a skin model. One study used a subcutaneous stretching device inflated over 7-days delivering a static stretching on the surrounding tissue [28]. A second study utilized an external stretching device glued to the dorsal midline to create 0%, 20%, 33%, and 40% of a static stretch [26]. Four studies corresponded to active stretching models. One of them includes both active and passive stretching paradigms [8,41–43].

**Rodent species.** Seventeen studies used a rat model of stretching. Of these, 14 studies stretched muscle [20–24,29–32,34–36,38,39]. In contrast, three studies used the skin of the lower back [28,41,42] Eight studies delivered their interventions using mice. Two stretched muscle [33,37] and six stretched the skin [8,25–27,40,43].

## Categories of inflammatory outcomes evaluated

Tables 3 and 4 summarize nine domains evaluated in the 28 studies: authors, study aims, type and intention of stretching, animal model information (strain, sex, species, age, N), intervention, stretching parameters, inflammatory outcomes, experimental groups, and results.

**Microscopic outcomes.** Twenty-two of the 25 studies used a microscopy technique to assess the effect of stretching on inflammation. Eighteen studies used histology, six used immunohistochemistry (IHC), two employed immunofluorescence (IF), one used second harmonic generation microscopy (SHG), and one terminal deoxynucleotidyl transferase dUTP nick end labeling (TUNEL) [21–37,39,40,42–44].

**Genetic outcomes.** Eleven studies explored changes in gene expression after stretching. Eight studies used RT-qPCR to assess gene expression of cytokines, growth factors, and fibrosis, two studies employed microarray analysis, and one study used RNA sequencing (RNAseq) [8,22,25–28,32,35,38,40,43].

**Cell/Protein/Lipid outcomes.** Three studies used flow cytometry to quantify inflammatory cells and cytokines [8,26,41]. One used liquid chromatography-tandem mass spectrometry to measure lipid mediators [41]. One study explored the effect of stretching on mesenchymal stem cells (MSCs) movement using an *in vivo* cell migration bioluminescent imaging model [28].

**Enzymatic/Protein outcomes.** Eleven studies used different enzyme-based immunoassay techniques to quantify protein- and lipid-mediators secretion after stretching. Eight studies used conventional ELISA. Two examined the secretion of specialized pro-resolving lipid mediators (SPMs). Seven measured cytokines and chemokines. Three used western blot (WB) for cytokines and muscle proteins. Two used conventional biochemical techniques to detect enzymatic activity [8,20,22,27,35,38–41,45,46].

**Macroscopic outcomes.** Three studies employed ultrasound to measure lesion size. One used a caliper for tumor size. One study injected Evans blue as a marker of vascular permeability, and another performed tensile testing of unwounded skin [8,20,27,41–43].

## Effects of stretching on inflammation

**Studies targeting the musculoskeletal system.** Of the 16 studies exploring the effect of stretching on different muscle groups, 15 used an shorter-term injurious passive stretching

protocol to cause a muscle inflammation [20,22,24,29–39]. The stretching protocols shared several characteristics and were relatively consistent across the studies (Table 3). Studies consistently found that muscle injury by stretching led to plasma extravasation, edema, necrosis, and myofiber degeneration. Six studies characterized cell infiltration [20,21,29,30,32,35,36,38,39]. Of these studies, two found a consistent increase of neutrophils and macrophages after an shorter-term bout of stretching [35,37], Less evidence exists regarding the upregulation and secretion of different cytokines and chemokines. Out of the five studies measuring cytokines and chemokines, three reported that IL-1b, TNF-a, and IL-6 increased after an shorter-term stretching protocol [20,35,38]. Of note, one study found that prior muscle stretching conditioning, using a protocol of lengthening or isometric contractions or passive stretches, was able to reduce inflammatory cell (neutrophils and macrophages) infiltration after subsequent (longer-term) delivery of the same stretching protocol.[37] This study's main finding was that these stretching protocols elevated neutrophil infiltration without causing overt signs of injury, suggesting that the inflammatory process itself may contribute to the induction of a protective mechanism [37]. A second study using a shorter-term stretching intervention observed that an inflammatory process was evident in young rats on day three after the intervention and returned to control values by day 10. However, these responses were not observed in the older rats, indicating that age blunted or muted the inflammatory process [38]. The only study exploring the longer-term effect of stretching in muscle found an increase in the cellular *interstitium* volume in the exposed limb of old rats suggesting inflammation (Table 3) [34].

**Studies targeting the integumentary system.** Nine studies explored the effect of stretching employing different longer-term stretching protocols [8,25–28,40,42,43,47]. These studies performed the inflammatory assessment at the end of the stretching intervention, or 12- to 24-hours post-intervention. Two studies explored the therapeutic effect of stretching on the lower back skin and connective tissue of rats using a carrageenan-induced inflammation model [41,42]. The first study established an active stretching model applied twice a day for 12 days to mitigate the carrageenan-induced inflammation. This study found stretching decreased lower back inflammation, leading to restored stride length and intrastep distance, decreased mechanical sensitivity of the back, and reduced macrophage expression in that area's nonspecialized connective tissue. The second study employed a similar carrageenan model; however, it included both shorter-term (twice a day for 48 hours) and longer-term (once a day for two weeks) stretching protocols, as well as active and passive stretching interventions [41]. They found stretching reduced inflammatory lesion thickness and neutrophil count and increased resolvin (RvD1) concentration within lesions [41,42]. A third study applied 10-minutes of active stretching once a day for four weeks on a p53/PTEN mouse mammary tumor cells model. The active stretching procedure was similar to the one used in the previous rat studies. Results showed that the stretched group had a 52% reduction in tumor volume, activation of the cytotoxic immune response, and increased SPMs [8,41,42]. A fourth study employed a mouse model of systemic sclerosis to test the same active stretching methodology. The animals were stretched once a day for 10 minutes/5 days a week for 4 weeks. Using ultrasound, they found, compared to control, stretching reduced skin thickness, increased subcutaneous tissue mobility, and downregulated genes related to systemic sclerosis in mice (*Ccl2* and *Adam8*) [43]. Nonetheless, this model of autoimmune inflammation did not show evidence that stretching attenuated inflammation. Across these four studies, 10 minutes of active stretching was enough to trigger localized inflammatory changes [8,41–43].

Five studies used a model of passive stretching delivered by different skin-stretching devices. The first study used an external device, allowing them to control for stretching duration and strain. Results showed that hair stem cells proliferate and hair regenerates when

**Table 3. Summary of effects of muscle stretching on inflammatory markers in animal models.**

| Authors | Study aim as described in the article | Type and intention of stretching | Strain, Sex, Species, age, N. | Intervention | Stretching parameters | Inflammatory outcome | Experimental groups | Results[#] |
|---|---|---|---|---|---|---|---|---|
| Baker et al. 2008 [29] | Quantify the acute phase myofiber response in young and old rats exposed to an acute bout of SSCs. | Passive Injurious model Pro-inflammatory study | Fischer Brown Norway hybrid male rats; 12 weeks old and 30-months old. N = 30 from each age. | • SSC • Shorter-term protocol | • Intensity (amplitude/ strain): 100% for 100 ms. • Duration: each set 10 SSCs repetitions. • Frequency: 8 sets/once. | • Histology: muscle edema • IHC: myosin heavy chain (MHCdev+) | • Randomized groups • Young and old rats randomized to SSCs groups with 6, 24, 48, 72, or 120 h recovery (N = 6 per time point). • No 0h recovery group | • Young and old rats displayed an increase in developmental myosin heavy chain (MHCdev+) labeling in the exposed muscle, indicating muscle regeneration. • Old rats displayed diminished MHCdev + labeling, compared with young rats, suggesting limited remodeling and (or) regenerative capacity. |
| Baker et al. 2006a [30] | Determine whether SSC muscle injury induces a temporal increase of myofiber degeneration, inflammation, and changes at the interstitial space. | Passive Injurious model Pro-inflammatory study | Sprague Dawley male rats. 12 weeks old. N = 72. | • SSC • Shorter-term protocol | Intensity (amplitude/ strain): 100% for 300 ms. Duration: each set 10 SSCs repetitions. Frequency: 15 sets/once. | • Histology: muscle fiber volume density and thickness /myofiber degeneration. | • Randomized groups • Experimental SSC group. • Isometric control group. • Randomized: 0.5, 6, 24, 48, 72, or 240 h recovery (N = 6 per time point). | • Increase in the volume density and average thickness of degenerating myofibers over time in the muscle exposed to SSCs that was greater than in muscle exposed to isometric contractions at 24, 48, and 72 h post-exposure. |
| Baker et al. 2007 [31] | Investigate the effect of repetitive SSC on rat skeletal muscle exposed *in vivo* using the left limb muscle groups. | Passive Injurious model Pro-inflammatory study | Sprague Dawley male rats. 12-wk old. N = 24. | • SSC • Shorter-term protocol | Intensity (amplitude/ strain): 100% for 100–300 ms. Duration: each set 10 SSCs repetitions. Frequency: 3, 7 or 10 sets/once. | • Histology: muscle inflammation • Stereology: myofiber evaluation | • Randomized groups • Isometric control group • 30-SSC group • 70- SSC group • 150-SSC group • 48h recovery | • A decrease in the percentage of volume density of normal myofibers in the 70- and 150-SSC groups. Percentage of volume density of degenerative myofibers and inflammation increased in the 70- and 150-SSC groups. • An increase in the percentage of volume density of degenerative myofibers in the 150-SSC group compared with the 70-SSC group was observed. |
| Baker et al. 2006b [32] | Study the changes in muscle morphology and measure changes in mechano-growth factor (MGF) gene expression in rat skeletal muscle exposed *in vivo* to SSC exercise of varying muscle lengths. | Passive Injurious model Pro-inflammatory study | Sprague Dawley male rats. 12 weeks old. N = 36. | • SSC • Shorter-term protocol | Intensity (amplitude/ strain): 100% for 100–300 ms. Duration: each set 10 SSCs repetitions. Frequency: 7 sets/once. | • Histology: myofiber degeneration • Stereology: thickness of normal and degenerative myofibers • RT-qPCR: MGF | • Randomized groups Isometric group 6h recovery • 48h recovery • Long and short muscle length injury groups 6h recovery. | • Exposure to SSC at longer muscle lengths results in greater morphometric indices of inflammation and a prolonged adaptation to SSC manifested by the lack of up-regulation in MGF mRNA. |

*(Continued)*

**Table 3.** (*Continued*)

| Authors | Study aim as described in the article | Type and intention of stretching | Strain, Sex, Species, age, N. | Intervention | Stretching parameters | Inflammatory outcome | Experimental groups | Results[#] |
|---|---|---|---|---|---|---|---|---|
| Brickson et al. 2014 [33] | Evaluate a standardized single stretch injury model to the biarticular gastrocnemius muscle using Achilles tendon (AT) shortening to control magnitude of injury. | Passive Injurious model Pro-inflammatory study | C57BL/67 male mice; 12 weeks old. N = 16. | • *In vivo* model of muscle stretch injury (shortening/lengthening contractions) • Shorter-term protocol | Intensity (amplitude/strain): 100%. Duration: each set lasted 2s. • Frequency: 2 set/once. | • Histology: Muscle fibrosis | • No mention of randomization • 4 mice per condition, 8 limbs in each group. • Sham control group or injury group according to one of three incremental levels of AT shortening • 24 h recovery | • Histological evaluation 24 h post-injury revealed increased morphological damage near the Musculotendinous junction (MTJ) in the AT shortened groups. • Damage was characterized by disruption of fibers and described in a semi-quantitative manner by counting multiple nuclei accumulation 24 h post-injury. |
| Cutlip et al. 2006 [34] | Investigate whether aging affects the ability of skeletal muscle to adapt to repetitive exposures of SSCs. | Passive Injurious model Pro-inflammatory study | Fischer Brown Norway hybrid male rats; 12 weeks and 30-months old. N = 11 | • SSC • Longer-term protocol | Intensity (amplitude/strain): 100% for 100–300ms. Duration: each set 10 SSCs repetitions/4.5 weeks Frequency: 8 sets/3 times/week. | • Histology: muscle characteristics | • No mention of randomization • Right muscle was the control. • Left muscle had 8 sets of 10 repetitions of SSCs • 24 h recovery | • An increase in the volume of the cellular interstitium was observed in the exposed limb of the old animals, which is indicative of an inflammatory response. |
| Dessem et al. 2010 [35] | Investigate muscle pain in the masticatory muscles because muscle tension is commonly associated with temporomandibular disorders and craniofacial pain. | Passive Injurious model Pro-inflammatory study | Sprague Dawley male rats. N = 198. | • Eccentric contractions (ECCs) • Shorter-term protocol | Intensity (amplitude/strain): 100% for 150ms. Duration: each set 100 ECCs. Frequency: 5 sets/once. | • Histology: myofiber membrane integrity. • IHC: neutrophil and macrophages infiltration • RT-qPCR: CGRP and $P2X_3$ • ELISA: cytokines | • No mention of randomization • ECCs group • CFA injection for comparison group • Contralateral muscle was used as controls for IHC • 24 h recovery | • Both EC and stretching disrupted myofibers produced plasma extravasation. IL-1β, TNF-α, IL-6, and vascular endothelial growth factor (VEGF) elevated in the masseter 24h following EC. At 48h, neutrophils increased and ED1 macrophages infiltrated myofibers while ED2 macrophages were abundant at 4d. • Muscle stretching produced hyperalgesia for 2d, whereas contraction alone produced no hyperalgesia. |

(*Continued*)

**Table 3.** (Continued)

| Authors | Study aim as described in the article | Type and intention of stretching | Strain, Sex, Species, age, N. | Intervention | Stretching parameters | Inflammatory outcome | Experimental groups | Results[#] |
|---------|----------------------------------------|----------------------------------|-------------------------------|--------------|----------------------|----------------------|---------------------|-----------|
| García et al. 2018 [21] | Evaluate alterations from different therapies in muscular injury using the Fractal Dimension (FD) method. | Passive Injurious model Pro-inflammatory study | Wistar male rats. 5-months old. N = 35. | • *In vivo* model of muscle stretch injury • Shorter-term protocol | Intensity (amplitude/ strain): 100%. Duration: each set 2s full contraction. Frequency: 10 sets/once. | • Histology: muscle inflammation • Fractal dimension: muscle inflammation | • Randomized groups • Divided into 5 groups: control; control injury; injury + ILT; injury + IP; injury + ILP. • 7 days recovery | • The groups submitted to the injury process demonstrated a process of inflammation, necrosis, and phagocytosis in the muscle. Myofibers with signs of stress were observed, such as polymorphic, rounded, and angular myofibers. |
| Geronilla et al. 2003 [36] | Introduce a novel method to evaluate, in real-time, changes in force parameters during injurious SSCs | Passive Injurious model Pro-inflammatory study | Sprague Dawley male rats. 12 weeks old. N = 24. | • SSC • Shorter-term protocol | Intensity (amplitude/ strain): 100% for 2.8s. Duration: each set 10 SSCs Frequency: 15 sets/once. | • Histology: muscle fiber degeneration and necrosis | • Randomized groups • Animals assigned to SSCs or isometric control group • 48 h recovery | • Histopathologic assessment of the tibialis anterior exposed to SSC cycles showed myofiber degeneration and necrosis with associated inflammation, while muscles exposed to isometric contractions showed no myofiber degeneration and necrosis, and limited inflammation. |
| Gluck et al. 2018 [24] | Assess the ability of SHG microscopy to visualize the extent of damage present by observing endoneurial collagen disruption throughout the healing process at various time points following acute stretch injury. | Passive Injurious model Pro-inflammatory study | Sprague Dawley female rats. 10-month old. N = 60. | • *In vivo* model of nerve stretch injury • Shorter-term protocol | Intensity (amplitude/ strain): • Low strain 14%—high strain 20% nerve maximum elongation. Duration: 5 min. Frequency: once | • SHG: collagen injury around muscle nerve • Histology: muscle inflammation • IHC: muscle inflammation | • Randomized groups • Divided into 6 groups per time after intervention. Day-0, 1, 3, 8, or 12-week recovery groups or control group. | • Low strain (LS) and high strain (HS) damaged nerves exhibit signs of structural collagen damage in comparison with sham control nerves. LS nerves exhibit signs of full regeneration, while HS nerves only partial regeneration with lasting damage and intra-neural scar formation. |
| Ozaki et al. 2015 [23] | Analyze changes in the skeletal muscle tissue of rats after muscle stretch injury (ECCs) using fractal analysis | Passive Injurious model Pro-inflammatory study | Wistar male rats. 5-month old. N = 21. | • *In vivo* model of muscle stretch injury (ECCs) • Shorter-term protocol | Intensity (amplitude/ strain): not provided. Duration: not provided. Frequency: 10 sets/once | Histology and fractal dimension (FD): muscle inflammation | • Randomized groups • Divided into 3 groups: Control (C); Lesion 2 days (L2) and Lesion 7 days recovery (L7) | The results showed high FD values of the inflammatory process in the experimental groups L2 and L7 in relation to control. The analysis of collagen in the picrosirius stained slides showed high FD in the L2 group compared to the L7. |

*(Continued)*

**Table 3.** (*Continued*)

| Authors | Study aim as described in the article | Type and intention of stretching | Strain, Sex, Species, age, N. | Intervention | Stretching parameters | Inflammatory outcome | Experimental groups | Results[#] |
|---|---|---|---|---|---|---|---|---|
| Pizza et al. 2002 [37] | Test the hypotheses that lengthening contractions, passive stretches, and isometric contractions increase muscle inflammatory cell concentration and that prior performance of lengthening contractions, isometric contractions, or passive stretches reduces muscle inflammatory cells after subsequent lengthening contractions. | Passive therapeutic stretching model Both/mixed Pro- and anti-inflammatory study | C57BL/6 male mice. 3- to 4-months old. N = 71. | • Lengthening contraction protocol • Shorter-term /longer-term protocol | Intensity (amplitude/strain): 20% for 5 min. Duration: each set 75 contraction repetitions. Frequency: 1 or 2 sets (separated by 2 weeks) | • IHC: neutrophils/macrophages muscle infiltration | • No mention of randomization • Single bout of lengthening, isometric contractions or passive stretches alone, or followed by a bout of lengthening contractions 2wk later. • Control groups normal cage activity + surgery • 3 h or 3 days recovery | • Three days after isometric contractions or passive stretches, neutrophils were elevated 3.7- and 5.5-fold, respectively, relative to controls. • Both macrophages and neutrophils were increased 51.2- and 7.9-fold, respectively, after lengthening contractions. • Prior lengthening contractions, isometric contractions, or passive stretches reduced inflammatory cells after lengthening contractions performed 2 wk later. |
| Rader et al. 2015 [38] | Characterize muscle fiber morphology 3- and 10-days following SSCs varying in repetition number (i.e., 0, 30, 80, and 150) for young and old rats. | Passive injurious stretching Both/mixed Pro- and anti-inflammatory study | Fischer Brown Norway hybrid male rats; 12 weeks old and 30-months old. N = 110. | • SSC • Shorter-term protocol | Intensity (amplitude/strain): 100% for 100–300 ms. Duration: set 30, 80 or 150 SSCs repetitions. Frequency: 1 set/once. | • Histology: muscle quantitative morphology • Microarray: gene expression • ELISA: cytokines | • Randomized groups • The contralateral muscle was the control • Animals assigned to 8 experimental groups for testing • 3- or 10-days recovery | • In young rats, muscle fiber degeneration was overt at 3 days after 80 or 150 SSCs and returned closer to control values by 10 days. For old rats, no such responses were observed. • Microarray analysis at 3 days: 2144 genes differentially expressed in young rats, while 47 genes in old rats. • Various cytokines and chemokines increased 3- to 50-fold following high-repetition SSCs for young rats with no change for old rats. |
| Ramos et al. 2018 [20] | Investigate the effects of low-level laser therapy on skeletal muscle strain (passive elongation) in an experimental model in rats. | Passive Injurious model Pro-inflammatory study | Wistar male rats. N = 210. | • *In vivo* model of muscle stretch injury (passive elongation) • Shorter-term protocol | Intensity (amplitude/strain): muscle elongation, 150% of the body mass. Duration: each set 20 min. Frequency: 2 sets/once | • *In vivo* injection: Evans blue extravasation marker of vascular permeability • Biochemical assay: CRP • ELISA: cytokines | • Randomized groups • 5 groups of six animals • Control: healthy rats • 0, 3, 6, 12, and 24 h recovery | • Plasma extravasation of groups treated with different doses of laser energy shows a reduction when compared with the stretch injury group. • Laser therapy reduced CRP and cytokine levels (TNF-α, IL-1β, IL-6, and IL-10). |

**Table 3.** (Continued)

| Authors | Study aim as described in the article | Type and intention of stretching | Strain, Sex, Species, age, N. | Intervention | Stretching parameters | Inflammatory outcome | Experimental groups | Results[#] |
|---|---|---|---|---|---|---|---|---|
| Smith et al. 2007 [22] | Provide evidence that TGF-b transcript and protein are induced in response to ECCs skeletal muscle injury. | Passive Injurious model Pro-inflammatory study | Sprague-Dawley (SD) female rats weighing approximately 225–250 g each. N = 11. | • *In vivo* model of muscle stretch injury (ECCs) • Shorter-term protocol | Intensity (amplitude/strain): not provided. Duration: each set 50 ECCs. Frequency: 1 set/once. | • IHC: TGF-b1 • RT-qPCR: TGF-b1 and 2. • WB: TGF-b1, TGF-b2 | • No mention of randomization • Contralateral muscle served as control • 48 h recovery | • Percentage of damaged myofibers was greater in the distal-most segment. • IHC revealed the presence of TGF-b1 in areas of myofiber injury, whereas TGF-b2 was not detected. • Increases in TGF-b1 and TGF-b2 precursor abundance were observed following strain injury. |
| Sudo & Kano, 2009 [39] | Hypothesize that the apoptosis response induced by ECCs would be activated in the regeneration phase as well as the inflammation phase. | Passive Injurious model Pro-inflammatory study | Wistar male rats 12 weeks of age. N = 27. | • *In vivo* model of muscle stretch injury (ECCs) • Shorter-term protocol | Intensity (amplitude/strain): 100% for 700 ms. Duration: each set 40 ECCs. Frequency: 1 set/once. | • Histology: muscle inflammation • TUNEL: myofiber apoptosis • WB: Bcl-2 and Bax | • Randomized groups • Each rat assigned to one of five groups 0, 1, 3, 7, or 14 days recovery | • At 1 and 3 days, focal edema and necrotic myofibers invaded by mononuclear phagocytes were present. Regenerated myofibers with central nuclei were detected at 7 and 14 days. The occurrence of TUNEL-positive myonuclei increased at 7 and 14 days compared with control. Myonuclear apoptosis was restricted to the subsarcolemmal space at 7 and 14 days and markedly absent from the central nucleus. The Bax/Bcl-2 ratio was higher at 3 and 7 days after ECC. |

SSCs, stretch-shortening contractions/cycles; ECCs, eccentric contractions; IHC, immunohistochemistry; IF, immunofluorescence; RTqPCR, quantitative reverse transcription polymerase chain reaction; MSC, mesenchymal stem cells; US, ultrasound; ELISA, Enzyme-Linked ImmunoSorbent Assay; ROS, reactive oxygen species; IL, interleukin; TUNEL, terminal deoxynucleotidyl transferase dUTP nick end labeling; WB, western blot; RNA-seq, RNA sequencing; SHG, second harmonic generation microscopy; FC, flow cytometry; SPMs, specialized pro-resolving mediators; *In vivo* BI, *in vivo* bioluminescent imaging; MGF, mechano-growth factor; FDS, *flexor digitorum superficialis*; FCR, *flexor carpi radialis*; SDF-1α, stromal-derived factor-1α; MIP-1α, macrophage inflammatory protein 1α; TARC, thymus and activation regulated chemokine; SLC, secondary lymphoid tissue chemokine; CTACK, cutaneous T-cell attracting chemokine; DMSO, dimethyl sulfoxide; PMNs, polymorphonuclear cells; ILT, injury and low-level laser therapy; IP, Injury and platelet rich plasma; ILP, injury plus low intensity laser therapy and platelet rich plasma. CGRP, Calcitonin gene-related peptide; P2X3, purinoceptor 3 gene; sclGvHD, murine model of systemic sclerosis.

# Results show main significant changes at P<0.05.

applying between 33% and 40% of strain for at least 7 days. This study also reports that macrophages are first recruited by chemokines produced by stretch and then polarized to the M2 phenotype. Subsequently, growth factors released by these macrophages activate stem cell proliferation and hair regeneration [26,28]. Likewise, another skin model surgically implanted a

**Table 4. Summary of effects of skin and surrounding connective tissue stretching on inflammatory markers in animal models.**

| Authors | Study aim as described in the article | Type and intention of stretching | Strain, Sex, Species, age, N. | Intervention | Stretching parameters | Inflammatory outcome | Experimental groups | Results[#] |
|---|---|---|---|---|---|---|---|---|
| Berrueta et al. 2018 [8] | Determine the effect of stretching on the growth of tumors implanted within locally stretched tissues in a mouse model of breast cancer. | Active therapeutic stretching model Both/mixed Pro- and anti-inflammatory study | FVB female mice. 6-week old. N = 66. | • Active stretching protocol • Longer-term protocol: 10 min once/day, for 4 weeks. | Intensity (amplitude/ strain): Not provided. Duration: 10 min Frequency: 10 min/day/ 4 weeks | • Caliper: tumor vol • FC: cytokines and cell infiltration. • ELISA SPMs: RvD1 and 2 • Microarray: gene expression. | • Randomized groups • Stretch group and control group • Assessment 24 h after last stretching | • Tumor volume at endpoint was 52% smaller in the stretch group, compared to the no-stretch group. • Results suggest a link between immune exhaustion, inflammation resolution and tumor growth. |
| Berrueta et al. 2016 [41] | Test whether stretching of connective tissue has a direct, local pro-resolution effect on tissue inflammation that can be monitored both in vivo and ex vivo. | Active and passive therapeutic stretching model Anti-inflammatory study | Wistar male rats. N = 27. | • Active and passive stretching protocols • Shorter-term (48h) and Longer-term (2wks) protocol | Intensity (amplitude/ strain): Not provided. Duration: 10 min Frequency: 10 min/once or twice a day/2 days or 2 weeks | • US: lesion size • FC: neutrophils • ELISA: SPMs • Lipidomic: SPMs | • Randomized groups • Active stretch group • Passive stretch group under anesthesia • Control group anesthesia alone • Assessment 12 h after last stretching | • Rats injected with carrageenan and randomized to stretch for 48 hours, stretching reduced inflammatory lesion thickness and neutrophil count, and increased Resolvin (RvD1) concentrations within lesions. |
| Chu et al. 2019 [26] | Design a specialized skin-stretching device that can identify how mechanical forces affect hair regeneration by modifying the strain. | Passive therapeutic stretching model Anti-inflammatory study | C57BL/6 and CCL2 null female mice. 6 animals per group. | • Skin-stretching device model • Longer-term protocol • Strain: 20, 33, 40%. Duration: 5, 7, 10 days. | Intensity (amplitude/ strain): 20, 33, 40% Duration: static/7 or 10 days. Frequency: once. | • RNAseq: gene expression • FC: macrophages infiltration (M1, M2) • RT-qPCR: cytokine • IF: skin | • Randomized groups • Groups were equipped with stretching device • Control groups were kept in the telogen phase. • Assessment at the end of each time point (5, 7, 10 days) | • Hair stem cells proliferate in response to stretch and hair regeneration occurs only when applying proper strain for an appropriate duration. • Macrophages are first recruited by chemokines produced by stretch and polarized to M2 phenotype. Growth factors such as HGF and IGF-1, released by M2 macrophages, then activate stem cells and facilitate hair regeneration. |
| Corey et al. 2012 [42] | Develop a novel model of non-specialized connective tissue inflammation and test that *in vivo* stretching of the back 2X/12 days improves gait, local inflammation and mechanical sensitivity. | Active therapeutic stretching model Anti-inflammatory study | Wistar male rats. N = 36. | • Active stretching protocol • Longer-term protocol 10'/ twice a day/12 day | Intensity (amplitude/ strain): Not provided. Duration: 10 min Frequency: 10 min/ twice a day/ 12-day | • US: Lesion • Histology: macrophages infiltration. | • Randomized by injection side and groups • Saline-no tto; saline-stretch; carrageenan-no tto; carrageen-sham; carrageenan-stretch. • Assessment 12 h after last stretching | • *In vivo* tissue stretch mitigated the inflammation-induced changes leading to restored stride length and intra-step distance, decreased mechanical sensitivity of the back and reduced macrophage expression in the nonspecialized connective tissues of the low back. |

*(Continued)*

**Table 4.** (Continued)

| Authors | Study aim as described in the article | Type and intention of stretching | Strain, Sex, Species, age, N. | Intervention | Stretching parameters | Inflammatory outcome | Experimental groups | Results[#] |
|---|---|---|---|---|---|---|---|---|
| Qiao et al. 2019 [25] | Hypothesize that mechanical stretching of the skin contributes to the pathogenesis of psoriasis by modulating keratinocyte function. | Passive injurious stretching model Pro-inflammatory study | Male BALB/c mice aged 8–10 weeks old. N = 15. | • Skin-stretching device model • Longer-term protocol 0.5ml/day/8 days H2O. | Intensity (amplitude/strain): not provided. Duration: static/8 days Frequency: Once | • Histology: skin inflammation • RT-qPCR: inflammatory genes • IF: skin inflammation | • No mention of randomization Group A silicone dilator • Group B dilator injection • Group C sham-operated • Assessment after day 8 | • Dilator-implanted mice displayed prominent epidermal hyperproliferation, impaired skin barrier function, and up-regulation of psoriasis-associated cytokines in epidermal keratinocytes. |
| Shan et al. 2017 [40] | Investigate the potential effect of naringenin on hypertrophic scar (HS) and its underlying mechanisms. | Passive injurious stretching model Pro-inflammatory study | KM female mice, 8-weeks old. N = 24. | • Skin-stretching device model • Longer-term protocol 10 days | Intensity (amplitude/strain): not provided. Duration: static/10 days Frequency: once | • Histology: skin inflammation • RT-qPCR: cytokine • WB: cytokines • ELISA: cytokines | • Randomized groups • Control group (A) 10% DMSO every day. • B and C models of HS• Assessment 24 h after last day of stretching | • Naringenin inhibited the formation of HS in a concentration-dependent manner. Naringenin inhibited fibroblast activation and inflammatory cell recruitment. mRNA and protein expression levels of TNF-α, IL-1β, IL-6 and TGF-β1 downregulated following naringenin treatment. |
| Wong et al. 2011 [27] | Detect transcriptional activity during scar formation and identified key inflammatory mechanotransduction pathways in skin fibrosis using genome wide microarray analysis. | Passive injurious stretching model Pro-inflammatory study | C57BL/6J female mice. 8–12 weeks old. N =? | • Skin-stretching device model • Longer-term protocol 10 days | Intensity: (amplitude/strain): 0.15–0.27 N/mm$^2$ (MPa). Duration: static/10 days. Frequency: once. | • Microarray: gene expression, fibrosis. • Skin tensile testing • IHC: inflammatory cells • ELISA: IL-4, IL-13, and MCP1. | • No mention of randomization • Experimental group: mechanical distraction device • Control group: device mounted not distracted. • Assessment 24 h after last day of stretching | • Scar formation in T-cell-deficient mice was reduced by almost 9-fold with attenuated epidermal and dermal proliferation. • Mechanical stimulation was highly associated with sustained T-cell-dependent Th2 cytokine (IL-4 and IL-13) and chemokine (MCP-1) signaling. • T-cell-deficient mice failed to recruit systemic inflammatory cells in response to mechanical loading. |
| Xiong et al. 2017 [43] | Determine whether in the absence of stretch, US measurement of skin thickness is increased, and subcutaneous tissue mobility are decreased in sclG-vHD. | Active therapeutic stretching model Anti-inflammatory study | Rag2-/-BALB/c and B10. D2 mice. N = 48. | • Active stretching model • Longer-term protocol • 10'/once a day/5 day a week/4 week | Intensity (amplitude/strain): Not provided. Duration: 10 min Frequency: 10 min/a day/5 days a week/4-week | • US: lesion size using ultrasound • Histology: assessment of the lesion • RT-qPCR: gene expression for extracellular matrix-associated pathway CCL2 and ADAM8 | • Randomized groups • 4 groups of mice • Syngeneic control/no stretch. • Syngeneic control /stretch • SclGvHD/no stretch • SclGvHD/ stretch • Assessment 24 h after last day of stretching | • Stretching reduced skin thickness and increased subcutaneous tissue mobility compared to no stretching at week 3. • Stretching also reduced expression of CCL2 and ADAM8 in the skin at week 4. Two genes known to be upregulated in both murine sclGvHD and the inflammatory subset of human SSc. No evidence that stretching attenuated inflammation at week 2. |

*(Continued)*

**Table 4.** (Continued)

| Authors | Study aim as described in the article | Type and intention of stretching | Strain, Sex, Species, age, N. | Intervention | Stretching parameters | Inflammatory outcome | Experimental groups | Results[#] |
|---|---|---|---|---|---|---|---|---|
| Zhou et al. 2013 [28] | Hypothesize that skin tissue undergoing mechanical stretch may synthesize and release a spectrum of cytokines that facilitate recruitment of circulating MSCs. | Passive therapeutic stretching model Both/mixed Pro- and anti-inflammatory study | Wild-type female Lewis rats. 4-weeks old. N = 12. | • Skin-stretching device model • Longer-term protocol inflation/ every other day/7-days | Intensity (amplitude/ strain): 60 mmHg/ pressure Duration: static/7 days Frequency: Once | • *In vivo* BI: stem cell migration • IF: skin inflammation • RT-qPCR: chemokines • *In vivo* MSC migration inhibition assay • Histology: skin inflammation | • Randomized groups • Twelve female Lewis rats into 2-groups: an expanded A and a Control B • *In vivo* assessment and at different time points with expanders (1, 4, 7, 14, and 21 days) | Expression levels of chemokines including MIP-1α, TARC/CCL17, SLC/CCL21, CTACK, and SDF-1α elevated in mechanically stretched tissues, as well as their chemokine receptors in MSC. Mechanical stretching induced temporal upregulation of chemokine expression. SDF-1a showed an increase in stretched skin, suggesting connection to migration of MSCs. |

SSCs, stretch-shortening contractions/cycles; ECCs, eccentric contractions; IHC, immunohistochemistry; IF, immunofluorescence; RTqPCR, quantitative reverse transcription polymerase chain reaction; MSC, mesenchymal stem cells; US, ultrasound; ELISA, Enzyme-Linked ImmunoSorbent Assay; ROS, reactive oxygen species; IL, interleukin; TUNEL, terminal deoxynucleotidyl transferase dUTP nick end labeling; WB, western blot; RNA-seq, RNA sequencing; SHG, second harmonic generation microscopy; FC, flow cytometry; SPMs, specialized pro-resolving mediators; *In vivo* BI, *in vivo* bioluminescent imaging; MGF, mechano-growth factor; FDS, *flexor digitorum superficialis*; FCR, *flexor carpi radialis*; SDF-1α, stromal-derived factor-1α; MIP-1α, macrophage inflammatory protein 1α; TARC, thymus and activation regulated chemokine; SLC, secondary lymphoid tissue chemokine; CTACK, cutaneous T-cell attracting chemokine; DMSO, dimethyl sulfoxide; PMNs, polymorphonuclear cells; ILT, injury and low-level laser therapy; IP, Injury and platelet rich plasma; ILP, injury plus low intensity laser therapy and platelet rich plasma. CGRP, Calcitonin gene-related peptide; P2X3, purinoceptor 3 gene; sclGvHD, murine model of systemic sclerosis; N/mm$^2$ = MPa Megapascal, a metric unit of pressure or stress, in terms of force per unit area.

\# Results show main significant changes at P<0.05.

silicone stretching expander to promote allogeneic luciferase-mesenchymal stem cells (Luc-MSCs) migration. Results suggested that the inflammatory process caused by stretching is responsible for the chemoattraction of MSCs [28]. By comparison, three passive stretching studies reported a pro-inflammatory effect of stretching. Two studies employed an established murine hypertrophic scar model using an external skin-stretching device for ten days, applying tension (0.27 until 0.96 N/mm$^2$), which triggered fibroblast proliferation and inflammatory cell recruitment [27,40]. A third study used a subcutaneous minidilator stretching implanter injected with up to 4 mL of H$_2$O (0.5 mL per day) to establish a mouse model of psoriasis. This procedure caused a prominent epidermal hyperproliferation, impairing skin barrier function, and upregulation of psoriasis-associated cytokines in epidermal keratinocytes [25]. Notably, all five studies mentioned the role of cytokines and chemokines in these inflammatory events [25–28,40]. However, only two studies indicated the relevance of monocytes infiltration and subsequent differentiation of macrophages (Table 4) [26,27].

## Discussion

This systematic review summarizes the findings of experimentally controlled rodent studies employing mechanical stretching forces on the musculoskeletal and integumentary (i.e., skin and surrounding connective tissue) systems to evaluate its potential impact on inflammatory

processes. To our knowledge, this is the first systematic review synthesizing the scientific body of literature on the impact of injurious and therapeutic stretching on inflammatory responses using experimental *in vivo* rodent models. The majority of identified studies focused on the impact of passive and shorter-term injurious stretching on muscle. Studies consistently controlled three key stretching parameters––stretching intensity, duration, and frequency––but not overall whole-body posture. A smaller number of studies evaluated the impact of active and passive therapeutic stretching on the integumentary system, with some attention to stretching intensity, duration, and frequency, but not body posture. Studies evaluating either the injurious and therapeutic effects of stretching reported a diversity of inflammatory outcomes, including macroscopic, histopathologic, genetic, immune cells sorting, cytokines, and protective lipids (e.g., resolvins). Few studies evaluated a long course of stretching (i.e., multiple sessions repeated over a period of weeks or months). Below, we discuss the potential translational relevance of these rodent model findings to human studies. We highlight how addressing current gaps in the pre-clinical *in vivo* literature might further inform clinical research on yogic stretching. We also discuss how a coordinated bi-directional translational research strategy, including both rodent and human studies, may be an effective approach for studying the effect of yogic stretching on inflammation and health.

## Current knowledge gaps between the isolated physical component of yoga (i.e., yogic stretching) and its interaction with systemic inflammatory, regenerative and remodeling processes

Yogic stretching primarily happens within the context of specific postures (called asanas) and during transitions between postures [48,49]. The impact of yogic stretching on specific body tissues can differ widely depending on the yoga style, teaching methods, the practitioner's experience and health status, and the duration, frequency, pace, intensity, sequence arrangement, and individual anatomical characteristics during each body posture [50]. These aspects of yogic stretching have received little to no research attention, hence the motivation of this review to mine rodent model stretching research.

One set of key physiological events believed to underlie the therapeutic effects of yogic stretching and conventional exercises are local and systemic immune-mediated processes, including inflammation, tissue remodeling, and regeneration occurring mainly in the myofascial and integumentary systems [51]. Recent systematic reviews evaluating the impact of yoga on inflammatory outcome suggest a link between yoga practice and inflammatory processes, however, these findings need to be interpreted carefully as the specific impact of the biomechanical processes associated with yogic stretching is typically embedded within a complex multimodal activity, including focused mental attention, breathing, imagery, and psychosocial interactions (e.g., group practice) which might also impact inflammation via alternative physiological pathways [7].

One systematic review summarized results from 15 studies assessing the long-term effects of yoga on inflammatory markers among healthy and/or disease individuals. Need a bit more here: One study of healthy participants found significant reduction in stimulated levels of IL-6, TNF-α, and IL-1β in ex vivo cultured blood in yoga group compared to controls [52]. However, another study of individuals reporting psychological distress found no significant reduction of IL-6, TNF-α, and CRP post intervention or at follow-up [53]. Overall, results showed no general agreement on the effect of yoga on levels of C-reactive protein (CRP), TNF-α, and IL-6 [54]. This should not be surprising given the heterogeneity of conditions, interventions, and time frames evaluated. Another earlier systematic review of randomized control trials

(RCT) on the effects of yoga on stress and mood reported similar findings with respect to inflammatory outcomes [55].

The available clinical research literature on yoga is limited in its ability to inform the specific effects of stretching on inflammatory biomarkers and clinical outcomes related to function and health [56]. This evidence gap highlights the value of summarizing findings related to design features and inflammatory outcomes in rodent stretching studies and their possible relevance to translational yogic stretching research. It is important to highlight that *in vivo* rodent stretching studies included in this review do not provide interventions that mimic the complexities of yoga practice. However, in both rodent stretching and human yoga studies, stretching events occur at the integumentary and myofascial systems, likely triggering inflammatory, regenerative, and remodeling processes. For these reasons we believe it is helpful to consider these studies together, and as part of a potential bi-directional translational approach to understanding the impact of yogic stretching on inflammation and health.

## Key protocol features of stretching paradigms

**Passive versus active stretching.** While yoga commonly utilizes both passive and active stretching, rodent studies identified for this review primarily employed passive stretching paradigms (89%; n = 22). To date, an agreed upon terminology for the types of stretching that spans humans and animals does not exist. In this systematic review, most studies considered active stretching as a dynamic activity in which the experimental rats or mice are trained and acclimated to be manipulated without any anesthetic protocol [42,57]. In contrast, passive stretching in rodent studies included: static mechanical forces employed either without the use of anesthetics (e.g., via an implanted stretching apparatus) or stretching while animals were anesthetized employing dynamic mechanical forces, for example during stretch-shortening contractions/cycles (i.e., SSC protocols including both concentric and eccentric stretching muscle contractions) [25,40,41] Stretching terminology used in human studies is primarily derived from the field of sports medicine and differs in meaningful ways [58]. For example, using sports medicine terminology, yogic stretching includes passive or static-passive stretching where the posture is held–for elongation–with support from some other part of the body or with the assistance of a partner or some other apparatus (i.e., props) between 10 seconds and less than one minute [59]. However, this passive stretching differs from rodent studies because humans purposely cooperate and are receptive (i.e., they try to relax the targeted area). Yogic stretching also includes 'static' or 'active-static' stretching (i.e., the body assumes a position and then holds it with the support of surrounding tissues, including the agonist muscles' strength) [2]. Furthermore, yogic stretching also utilizes dynamic stretching during flow sequences, which involve moving parts of the body and gradually increasing reach, speed of movement, or both [60]. Inside this dynamic category, specific yogic stretching sequences can be considered ballistic, active, resistance, and loaded stretching modalities, all occurring, for example, during a widely used sequence call the "sun salutation." A growing body of research in physical therapy and sports medicine supports that both passive and active stretching in humans show clinical benefits (e.g., enhancing function in multiple musculoskeletal pain conditions) [61–63]. Less research has evaluated the impact of these stretching techniques on inflammatory processes [28].

There are several advantages to the use of passive stretching protocols in rodent studies. They provide more precise control over several aspects of the stretching procedure, such as stretching intensity, duration, and specific localization, and eliminate the role of top-down cognitive/affective contributions to stretching. Several studies identified in this review performed passive stretching under anesthesia to eliminate variability in forces, as well as

cognitive /affective contribution [25,40,41]. For example, studies by Baker et al. precisely applied different intensity and frequency of stretch-shortening contractions (SSCs) to deliver an injurious stretching intervention in muscles, which led to muscle inflammation and myofibers degeneration, mainly in older rats [29–32]. Others, such as Chu et al. used an external device to stretch the skin and surrounding connective tissue and found that stretching promotes hair stem cell proliferation and regeneration, driven by macrophages recruited by inflammatory chemokines and M2 polarization [26]. However, it should be noted that findings from passive rodent stretching models obviously do not reflect the impact of the more complex yogic stretching, which through the use of precise overall body postures, integrates forces from other body segments. To better inform yoga-related research, future passive stretching animal studies should include stretching protocols with postures that simulate whole body yogic postures, such as the those used in active stretching rats, mice and pigs paradigms [8,41,57]. These studies should also give more consideration to control for potential animal's stresses associated with handling and/or anesthesia, and/or provide estimates of the magnitude of these stressors [26,33,40].

A small proportion of studies in this review (14%, n = 4) utilized active stretching techniques and the majority of these evaluated therapeutic effects on inflammatory processes. For example, Berrueta et al., Corey et al., and Xiong et al. actively trained mice and rats to be lifted by the tail to engage in whole body stretching, thus including different muscle groups, skin, and connective tissue [8,41–43]. This experimental stretching protocol partially simulates postures used in yogic stretching, such as backbends and inversions [64]. These rodent studies used a Carrageenan model of subcutaneous inflammation and found that stretching reduces lesion size, neutrophil recruitment, and increased secretion of protective lipids (e.g., resolvins). More recently, Vergara et al. adopted a wheelbarrow active stretching paradigm in a porcine model, using a similar carrageenan-subcutaneous inflammation model. They reported that pigs learned and tolerated the active stretching procedure well and that compared to a control group, the average lesion area was significantly smaller in the stretching group [57]. Collectively, these studies suggest some form of active stretching, partially mimicking yogic stretching, could impact inflammatory, remodeling and regenerative processes commonly present in different inflammatory diseases or after a programmed tissue injury (e.g., surgery) [8].

Compared with passive stretching, rodent studies using active stretching paradigms are more likely to inform stretching that takes place during yoga training. As in passive stretching, future animal research should develop novel active stretching paradigms that better simulate the diversity of postures typically encountered in yogic stretching. Consideration should also be given to the stresses introduced to animals due to human contact during active stretching paradigms [42,43,57]. A possible alternative to manually imposed animal stretching would be to develop models of active self-stretching. For example, the magnitude, intensity, and frequency of active whole-body stretching could be achieved by using physical structures introduced to the animal's environment to promote specific postures and movements (e.g., tube-mazes and running wheels of different dimensions, strategic placement of water, or hidden food treats) [65,66]. Wearable devices could also be deployed. For example, Langevin et al. and Bishop et al. utilized harnesses that systematically controlled the magnitudes of trunk and limb mobility in pigs to study wound healing after a fascia injury in the dorsal truck at the L3-4 vertebral level [67,68]. Parallel studies could be designed in humans, with an aim to either minimize or enhance short-term mobility (e.g., kinesio taping—a technique widely used in sports medicine to restrict movements around joints and soft tissues) [69].

**Stretching parameters (intensity, duration, frequency, and posture) and stretching protocol duration (shorter- vs. longer-term).** In addition to the active vs. passive nature of stretching, key parameters such as intensity, duration, frequency, and overall body posture are

likely to influence to the impact of stretching. Clinical yoga research protocols have begun to systematically control the types of asanas (i.e., postures) used, the postures' holding duration, and the practice frequency to inform therapeutic effect [70]. However, clinical studies have not yet systematically controlled the intensity of stretching or the impact of specific postures on local or systematic biological or clinical outcomes (e.g., inflammation and chronic pain) [17].

The rodent studies included in this review report a variety of experimentally controlled stretching protocols accounting for three key parameters: intensity, duration, and frequency––parameters that are appealing because of their quantitative nature [17]. The majority of these studies employed shorter-term stretching interventions with a focus on injury. For example, with respect to an injurious intervention, Baker et al. precisely controlled three stretching parameters during the stretch-shortening contractions (SSCs) of the tibialis anterior muscle in rats [29–32]. As an example of a therapeutic intervention, Pizza et al. controlled intensity (20% amplitude/strain), duration (each set of stretching included 75 repetitions), and frequency (1 or 2 sets separated by 2-weeks) and reported that acute isometric contractions or passive stretches elevated neutrophils, whereas a longer-term intervention, separated by two weeks, showed a therapeutic effect by reducing neutrophil infiltration [37].

Ten of 25 studies employed carefully controlled longer-term stretching protocols. Nine of these stretched the integumentary system with a focus on both therapeutic and injurious outcomes. For example, Berrueta et al. actively stretched female mice 10 minutes (i.e., duration) daily for four weeks (i.e., frequency) using the previously mentioned tail lifting paradigm. They reported tumor volume reduction (52%), higher cytotoxic immune response, and elevated levels of protective lipids (i.e., resolvins) in the stretch compared to the no-stretch group [8]. Of note, Berrueta et al., Corey et al., and Xiong et al. controlled two stretching parameters (duration and frequency) quantitatively. However, posture was only controlled qualitatively, and they did not control for stretching intensity [8,41–43]. The remaining longer-term stretching studies did not stretch the whole animal body, targeting only a portion of their skin and surrounding connective tissue. For example, Quiao et al. used subcutaneous devices for continuous and static skin stretching (i.e., duration) for eight days to explore its injurious effects. They only controlled for stretching duration and frequency, reporting epidermal hyperproliferation, impaired skin barrier function, and up-regulation of psoriasis-associated cytokines in epidermal keratinocytes [25]. Although studies using skin stretching devices do not directly link to yoga interventions, these studies offer insights into basic mechanisms associated with mechanotransduction, wound healing, and tissue regeneration processes, which might more broadly inform fundamental biological mechanisms associated with yogic stretching [26,71].

Regarding the posture parameter, the ability to adopt precise body posture during yogic stretching practice is crucial [72]. Of note, none of the rodent studies included in this systematic review objectively accounted for overall whole-body posture during stretching of body segments. A few studies provided a qualitative explanation of the animals' position. For example, Brickson et al. placed the rats lying side-wise to manipulate the left hind limb [33].

Future animal studies should emphasize longer-term interventions, their potential therapeutic effects and assume better control of stretching intensity, especially with protocols stretching the whole animal's body since none of the studies included here designed a model with these components [8,41–43]. Laboratory-based studies could utilize widely available kinematics and motion detection technology to objectively and quantitatively characterize posture and magnitudes of tissue displacement [73]. These technologies would also permit the evaluation of how different stretching postures impact the activity of specific muscle groups and tensile properties of connective tissue [74]. Parallel experimental human studies could rely on wearable sensors or kinematic systems capable of measuring tissue displacement, joint angles, and symmetry, as well as employ protocols that mechanically and systematically deliver a

more precise magnitude of forces targeting specific body parts (e.g., Cox chiropractic table to control for flexion/distraction in the thoracolumbar region) [75].

**Injurious versus therapeutic stretching.** While the primary focus of clinical yoga research to date has been to evaluate its therapeutic effects [55], a growing body of studies has begun to evaluate its injurious side effects systematically [76]. Specific postures, such as hand-, shoulder- and headstands, have been associated with adverse events, with the most commonly reported being muscle or joint pain, soreness, and strains. Adverse events appear to be higher among participants with chronic diseases and self-study practitioners without specialized supervision [77]. These injuries are typically minor and affect mainly muscle groups and the surrounding connective tissue, including fascia [76,78]. Findings from injurious stretching interventions in animal studies may inform strategies for minimizing such adverse events by improving our understanding of the systemic and localized effects of yogic stretching-related injuries in humans [79,80]. Currently, muscle injurious animal studies identified in this review were motivated by stretching issues specific to sports medicine and rheumatology. They typically studied the impact of passive elongation and stretch-shortening cycles (SSCs) on muscle injury. For example, Rader et al. used an injurious SSCs protocol to compare muscle fiber changes in young and old rats, and Ramos et al. used passive elongation in an experimental rat model of muscle strain [20,38]. These controlled muscle studies offer several advantages. First, they allow evaluation of tissue adaptation using microscopic techniques, which are less feasible in human studies. Second, they enable *ex-vivo* or *in-vitro* studies to focus on the effect of stretching on specific tissues or cell types. Third, they offer the possibility for controlling different stretching parameters (i.e., duration, frequency, and intensity). Of note, no injurious studies evaluated the impact of stretching on inflammation resolution, a key factor in injury recovery. Future animal studies could be designed to focus on the most common types of injuries observed in yoga clinical trials.

Only seven of the 25 studies included in this review specifically evaluated the potential therapeutic effect of stretching [8,26,28,37,41–43]. In all cases, stretching regimens indicated positive effects with respect to inflammation-related outcomes. Of those seven studies, four studies utilized active stretching [8,41–43], three evaluated passive stretching protocols [26,28,37], and six out of seven utilized long-term interventions [8,26,28,41–43]. The tissues targeted included muscle, skin, and connective tissue. The inflammatory outcomes ranged from immune cell infiltration, cytokines, resolvins, gene expression, and lesion size (Tables 3 and 4). The small number of studies and the heterogeneity across them make it premature to draw general conclusions about the therapeutic effect of stretching on inflammation-related outcomes. However, it is noteworthy that studies employing the whole body in active stretching protocols for an extended period of time, showed a positive effect. More generally, these studies support the use of rodent models to simulate some aspects of yoga-like stretching.

Hypothesized mechanisms underlying observed therapeutic effects varied with study design. For example, Pizza et al. suggested that after stretching, inflammatory events afford protection from contraction-induced muscle injury through inflammatory protective mechanisms (e.g., the relationship between stretching and immune cell infiltration) [37]. Chu et al. suggested that precise control of stretching parameters is involved in hair regeneration through the alternative activation of macrophages [26]. Zhou et al. suggested that the therapeutic effect of mechanical stretching on the skin and connective tissue can induce the release of specific cytokines that facilitate stem cell migration involved in tissue repair and regeneration [28]. Compared with human theories that suggest stretching elicits its benefits by promoting proper muscle and connective tissue function [81], these rodent studies targeted specific biological extracellular pathways. Therefore, these *in vivo* studies further support the value of bidirectional research on unveiling similar mechanisms among humans. Future animal studies

might expand upon investigations evaluating stretching protocols with therapeutic intention. Whereas in human studies, there is a need for well controlled studies evaluating the impact of yogic-stretching on inflammatory processes, including those associated with acute and chronic musculoskeletal pain, to inform rehabilitation and prevention. One specific opportunity would be to evaluate the therapeutic benefits of stretching delivered perioperatively for surgical procedures, to test if stretching can reduce the risk of postsurgical chronic pain and long-term use of opioids [82]. Provocative preliminary results indicating that stretching can reduce the rates of tumor growth also suggest that well designed human stretching studies should go beyond studying the impact of yogic stretching on symptoms and quality of life in cancer patient [83], and begin to explore mechanotransduction processes underlying cancer progression [54].

**Inflammatory outcomes used in rodent stretching studies.** Studies included in this review evaluated a wide range of inflammatory outcomes. The most consistent inflammatory outcomes among the injurious muscle studies were histopathological changes (i.e., plasma extravasation, edema, necrosis, and myofibers degeneration), confirming the acute injurious effect of stretching [20–22,33,35–39]. Some exceptions include Gluck et al. [24], who studied histopathological changes after stretching the nerve innervating specific muscles, and Pizza et al. [37], who utilized immunohistochemistry (IHC) to quantify histopathological changes in neutrophil and macrophage populations during shorter- or longer-term stretching protocols. Of note, this group of studies did not describe histological changes in the connective tissue and fascia surrounding muscle groups. While histopathology is a standard laboratory technique used in basic science, its translation into a clinical trial is less common and feasible. Instead, clinical researchers used other mechanisms to explore muscle changes, such as ultrasound, dynamometers, and electromyographic activity (EMG) [84].

Fewer rodent studies reported inflammatory outcomes using inflammatory markers (e.g., cytokines), immune cell infiltration, and gene expression within the muscle [20,35,38]. As noted above, there was no mention of protective lipids measurements (e.g., resolvins). Non-invasive outcomes, similar to those used in clinical studies, such as dynamic ultrasound imaging, were not used in animal studies to evaluate age-related changes in myofibers, connective tissue, and fascia after stretching [85,86].

To date, no yoga studies have explored the local effect of isolated stretching on inflammatory processes using any of the techniques mentioned above. Clinical studies exploring the effect of yoga on systemic levels of cytokines have found evidence, for example, in decreased levels of IL-6 and TNF-α and increase adiponectin [87]. Of note, these findings are different from a large body of sports studies suggesting acute exercise results in a transient increase of IL-6, IL-10, IL-8, IL-18, and IL-1ra [88].

In the subset of rodent studies, in which stretching targeted the integumentary system, 7 out of 9 studies performed histopathological, immunofluorescence, or IHC analysis to describe microscopic changes in the skin subjected to longer-term protocols of stretching [25–28,40,42,43]. As noted above in studies targeting muscle, the use of these techniques in clinical trials poses challenges; however, *in vitro* studies using resident human dermal fibroblast for stretching could be one way to overcome the limitations of histopathology [89]. One critical need to inform the translation between animal and human studies is the development of human experimental models to evaluate the local impact of stretching at the site of injuries or lesions. One example of such an approach would be to intradermally or subcutaneously inject a low dose of endotoxin (i.e., lipopolysaccharide—LPS) to trigger a focal inflammatory process, in which a precise skin stretching modality could be applied. Later, microbiopsies might help unravel changes in the subcutaneous stroma (i.e., collagen, fibroblast, and other stromal cells).

These models could inform surgeons about the future repair and regenerative processes involve after surgery.

Another approach to quantify the effect of stretching on inflammation resolution was introduced by a subgroup of rodent studies using active stretching paradigms to study lesion size, measured by ultrasound, or tumor volume using a caliper, in rats and mice, respectively. These studies, after animal euthanasia, also processed different biological samples for histology, gene expression, cell sorting, and protective lipids detection (lipidomic and ELISA) [8,41–43]. The feasibility of using non-invasive assessment of inflammation, such as high-resolution ultrasound, to evaluate stretching's local effect is appealing. A study lead by Ellis et al. employed ultrasound elastographic measurements to assess the feasibility and reliability of this technology to quantify shear strain at the sciatic nerve-hamstring muscle interface during active and passive knee extension-flexion movements in healthy people [90]. Although they did not access inflammation, this model could be extended towards musculoskeletal chronic inflammatory pain conditions. Langevin et al. have used ultrasound to evaluate the effect of stretching in a porcine model of thoracolumbar fascia movement restriction [67,68].

**Animal species.**   The rat paradigm was the primary model used to examine the effect of stretching on muscle or muscle innervation. Conversely, those studies stretching the skin/connective tissue relied on the mouse model because it allows them to increase the number of animals per group and use continuous static stretching devices underneath or on top of the skin [8,25,26,40,43,91]. Animal studies using models that closely resemble human anatomy (e.g., pigs, dogs, primates) are substantially more expensive and complex to execute [92]. These animal models might allow active self-stretching paradigms by proper training, eliminating significant stressors, such as the anesthetic protocol and human handling. Recent studies evaluating stretching in a pig model provide examples in which active stretching was used to evaluate changes in the thoracolumbar fascia, along with functional changes in fascia mobility and gait speed [57,67,68].

## Strengths and limitations

This study has a number of noteworthy methodological strengths. First, it followed the Systematic Review Centre for Laboratory Animal Experimentation guidelines (i.e., the S1 Protocol format for systematic reviews of animal intervention studies). Second, data extraction and methodological assessment involved multiple reviewers. Third, it assessed the quality of each study using the SYRCLE risk of bias tool [15,16].

This review also has multiple limitations. First, it did not include *ex vivo* or *in vitro* studies exploring changes in cell proliferation, elongation, migration, apoptosis, or necrosis due to stretching [89]. These studies have advantages over the *in vivo* models because they allow even more control and precision during the stretching intervention (i.e., duration, frequency, and intensity). However, *ex vivo* or *in vitro* studies' relevance to informing the design of clinical studies evaluating yogic-like stretching is quite limited because the natural microenvironment in which tissues and cells are located in the whole animal body is disrupted. Second, while the search strategy for this review employed a range of terms associated with stretching (e.g., active stretching, passive stretching, stretch injury, mechanical stretch), this strategy may have excluded studies evaluating the impact of mechanical forces induced by only eccentric contractions, mobilization techniques, manual physical therapy, massage, exercise or longer-term repetitive strain injury impact inflammatory outcomes [74,93]. Third, this systematic review did not include search terms targeting delayed onset muscle soreness (DOMS), which may have excluded studies evaluating the impact of physical stretching on muscle soreness and inflammation using rodent models [94]. Fourth, this systematic review only included studies

written in English. Finally, while this study largely adhered to the S1 Protocol format for systematic reviews of animal intervention studies and SYRCLE risk of bias tool, the final search strategy was not approved by a professional librarian's.

## Supporting information

**S1 Table. Electronic search strategy.**
(DOCX)

**S1 Protocol.**
(DOCX)

## Author Contributions

**Conceptualization:** Dennis Muñoz-Vergara, Peter M. Wayne.

**Data curation:** Dennis Muñoz-Vergara, Weronika Grabowska.

**Formal analysis:** Dennis Muñoz-Vergara, Weronika Grabowska.

**Funding acquisition:** Dennis Muñoz-Vergara, Peter M. Wayne.

**Methodology:** Dennis Muñoz-Vergara, Gloria Y. Yeh, Peter M. Wayne.

**Project administration:** Dennis Muñoz-Vergara.

**Supervision:** Gloria Y. Yeh, Peter M. Wayne.

**Validation:** Sat Bir Khalsa, Kristin L. Schreiber, Christene A. Huang, Ann Marie Zavacki.

**Writing – original draft:** Dennis Muñoz-Vergara.

**Writing – review & editing:** Dennis Muñoz-Vergara, Sat Bir Khalsa, Kristin L. Schreiber, Christene A. Huang, Ann Marie Zavacki, Peter M. Wayne.

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
