## [Decision Letter · Decision Letter 0]

2 Feb 2022

PONE-D-21-20626A systematic review of in vivo animal stretching regimens on inflammation and its relevance to yoga researchPLOS ONE

Dear Dr. Munoz Vergara,

Thank you for submitting your manuscript to PLOS ONE. After careful consideration, we feel that it has merit but does not fully meet PLOS ONE’s publication criteria as it currently stands. Therefore, we invite you to submit a revised version of the manuscript that addresses the points raised during the review process.

The manuscript has been evaluated by four reviewers, and their comments are available below.

The reviewers have raised a number of concerns that need attention. Could you please revise the manuscript to carefully address the concerns raised?

We look forward to receiving your revised manuscript.

Kind regards,

Sebastian Shepherd

Associate Editor

PLOS ONE

Journal Requirements:

Reviewers' comments:

Reviewer's Responses to Questions

**Comments to the Author**

1. Is the manuscript technically sound, and do the data support the conclusions?

Reviewer #1: Yes

Reviewer #2: Partly

Reviewer #3: No

Reviewer #4: Yes

2. Has the statistical analysis been performed appropriately and rigorously? 

Reviewer #1: Yes

Reviewer #2: N/A

Reviewer #3: No

Reviewer #4: N/A

3. Have the authors made all data underlying the findings in their manuscript fully available?

Reviewer #1: Yes

Reviewer #2: No

Reviewer #3: Yes

Reviewer #4: Yes

4. Is the manuscript presented in an intelligible fashion and written in standard English?

Reviewer #1: Yes

Reviewer #2: Yes

Reviewer #3: Yes

Reviewer #4: Yes

5. Review Comments to the Author

Reviewer #1: Thank you for this excellent manuscript. Please consider the following minor editing/improvement suggestions:

170 Table 3: Both of the first two stretching terms include “movement applied by an external force to increase range of motion” within their definitions. Would it be possible to specify more clearly the differences between these terms in this respect; i.e when such an externally induced tissue stretch will be counted as ‘active’ and when as ‘passive’?

170 Table 3: The inclusion of the term ‘In vivo model of an acute nerve stretch injury’ in the left column could possibly be seen as a contradiction to the previously specified inclusion of studies evaluation “muscles, skin and connective tissue” only. Since the nerve bundles evaluated are often part of a larger ‘muscle’ or ‘connective tissue’, their inclusion could therefore be possibly justified based on this or similar reasoning. Please clarify your reasons for inclusion.

644 I suggest including a brief discussion of potential age effects within this larger ‘DISCUSSION’ section. You Already ADDRESSED THIS ASPECT IN LINES 294-297. At least in humans, there are recent indications that in older individuals the stretching effect may primarily target different tissues than in younger ones (see e.g. doi: 10.1249/MSS.0000000000002360). If verified, this could possible have some interesting implications for yoga practice.

Above all: my congratulations to this very good review study and manuscript.

Reviewer #2: The present study aimed to inform a basic biological framework for designing and interpreting clinical studies that evaluate the impact of yogic stretching on inflammation and health. This is an interesting study, however, I have some major concerns in procedures.

1. First, the novelty of the results/analysis is poor.

2. I wonder if this manuscript is not aligned with guidelines or poor description, although the authors cited [de Vries, RBM, Evidence-based Preclinical Medicine, 2015], the guidelines of the Systematic Review Center for Laboratory Animal Experimentation. The authors can follow PRISMA and SYRCLE. That is through inclusion/exclusion criteria, search strategy, data extraction, and risk of bias.

3. The inclusion/exclusion should be defined more clearly, based on PICO even if systematic review for animal models. Is the exclusion criteria only “all original peer-reviewed articles”? For example, lung stretch? Vascular stretch? Meanwhile, the authors showed some reason for exclusion in Fig 1.

4. I am not sure of the validity of the search strategy. (1) The therapeutic stretching technique is sometimes included in mobilization, manipulation, manual therapy, and so on. Why didn’t the authors include the term? (2) Why were the injurious stretching techniques different from delayed onset muscle soreness model, or eccentric muscle contraction model? Table 3 is not enough to show that. (3) Can the authors justify the species as only rat or mouse, not other animals in Table 1? Meanwhile, the results included the articles in rabbit. (4) Why the term for yoga is used “AND search” in supplement 1? The treatment should be stretching. (5) Didn’t the authors communicate with the librarian? (6) Didn’t the authors submit PROSPERO before this study? (7) The periods of search should be updated whenever.

5. The variables for data extraction were shown. However, the procedure of actual data extraction is not shown. Multiple inflammatory outcomes? Multiple time points? Mean and SD? How about discrepancies between reviewers?

6. Could the authors justify the modified checklist for risk of bias? SYRCLE's risk of bias tool for animal studies is also published based on Cochrane recommendations.

7. This manuscript of systematic review is in vivo animal stretching studies, not yoga research. The title would cause misunderstanding.

Reviewer #3: The authors describe a systematic review of reasonable quality on animal models for stretching. As a great fan of systematic reviews on animal models, I was very enthusiastic about this paper upon reading the title in the review request. Upon reviewing the paper however, my enthusiasm progressively decreased. The methods leave room for improvement (see below), there is relatively little real data synthesis (the authors repeat the findings from the included papers but hardly add to them except for several suggestive vote counting exercises), referencing is tremendously liberal, and some of the conclusions are extremely far-fetched.

As I appreciate the effort that has gone in to collecting the data, I do provide a full review report, hoping that with a thorough rewrite the paper may become acceptable for publication. The data are certainly interesting.

Major issues:

The relevance of the animal models to yoga remains completely unclear. The animal models have very little resemblance to voluntary conscious stretching up to a certain “sweet spot”. The face, concept and predictive validity of the animal models for yoga is not at all discussed. Without a comparison with human data, you cannot state anything on the topic (other than that it inspired the review perhaps). I suggest the authors read into the topics of animal model validation and animal-to-human translation, and either leave the yoga aspect out of the paper, or include human data for comparison (which could make this paper my favorite read of the year, and I do dare to say that in January).

Study quality scores are highly disputable. They are not very relevant, high scores can go together with low risk of bias and vice versa. Please skip these, and provide table 2 totals by column instead of by row, this is far more interesting.

Definitions for active and passive stretching overlap. This needs to be clarified. Without clarification figure 2A is meaningless.

Table 4 and 5 are not internally consistent and need harmonization. An example: “no mention of randomisation” for Best et al., but not for the other non-randomising studies. Also, why the bold? Can the name of the second column be changed to “study aim as described in the paper” (or something similar)? Is there a reason to the bold text? I find it confusing. If there is a consistent decision making process behind the inconsistencies, please describe it somewhere beforehand.

In the results section, there is quite a bit of phrasing that encourages vote counting (e.g. “of these [six] studies, five found …”. If there are 6 studies, a meta-analysis would be interesting. Without effect sizes, 5 out of 6 sounds informative but is not (which makes the phrasing misleading).

The authors refer to an evidence gap in human literature. I think the human literature needs to be further described (what were the comparators in the 25 randomised control studies included in the systematic review that is reference 73? and in the 301 randomized controlled trials of yoga summarized in reference 121? Is there really nothing that can be analysed there?). I also think resolving this human evidence gap should be prioritized over further animal studies because a. humans can give informed consent, and b. the value of the animal models has not been proven. Remove all suggestions for future animal experiments from the manuscript.

Minor comments:

L30: were studies <1900 not eligible or is this the general database cut-off?

L70-71: if the authors want to state that inflammation is evolutionary preserved, this needs a specific reference. I think it will be difficult to find one, as the immune system varies substantially between species and we do not fully understand how.

L76: “biological framework”? Please explain or leave out.

L91: delete “structure and”, and change “guidelines” into “format” or “template”. Also, why was the protocol not posted?

L101 and further: why are the species not included in all searches? Why are some species searched for and others not? Why were the available animal filters not used? (doi: 10.1258/la.2010.009117 and 10.1258/la.2011.011087, and for future work 10.1177/00236772211045485)

L108: delete “different” (current phrasing suggests that you only included studies that compared multiple stretching techniques, and excluded stretching vs. control only)

L122: did these reviewers evaluate all retrieved studies independently or did both do half the set? Was this the same for all phases?

L137: Why not the SYRCLE Risk of Bias tool?

L158: change “included” into “described”. For most animal studies a power analysis is performed, as it is a mandatory part of the ethics review process.

L363: I think you mean “either” where you write “both”.

L403: Please change “one” into “four”. (Yin-yoga is gaining popularity.)

L625 and elsewhere: please delete figure 4. It has little to do with your evidence which focussed on in vivo (in rodents) only.

Paper flow chart: please move the box with the duplicates to where it belongs (they go out, not through)

Reviewer #4: The authors have done great job in preparing and writing this systematic review article, including 1,411 English articles between 1900 and 2020 as per their ‘predefined criteria’.

This systematic review summarizes the findings from 28 experimentally controlled animal studies (in vivo) conducted in last 20 to 25 years, employing mechanical stretching forces on the musculoskeletal and integumentary (i.e., skin and surrounding connective tissue) systems to evaluate its potential impact on inflammatory processes, and its relevance to yoga research.

It may be an effective and useful approach for studying the effect of yogic stretching on inflammation and health as well as future translational research.

There are some queries/suggestions which may be addressed by the authors and may add more value to this article:

1. The main concern is that yoga includes “active” stretching exercises (asana) in humans, whereas authors have used 25 (out of total 28) animal studies using “passive” stretching exercises. How it can be justified for this systematic review where theses “passive” animal studies are being tried to relate with “active” yoga stretching exercises (asana)?

The authors try to explain this phenomenon using concept of ‘stretching’ as per sports medicine terminology, that does not appear much convincing, scientifically.

2. The animals used in these studies were primarily rats or, mice. Again, how this stretching exercises in tiny rats/mice can be compared with human yoga stretching exercises (asana)?

3. Why authors have included the studies using stretching of skin of lower back (3 studies out of total 28) to understand the related pathophysiology. The studies using skin may be removed from the list/ table 5.

4. In the animal stretching (under anesthesia or, without anesthesia) vs human yoga stretching (asana), how the role of cognition and stress can be ignored?

The stress during these stretching exercises is different in humans vs tiny animals (rats/mice).

5. Page 4, line 53: Can we delete the word, “therapeutic” from here. As the yoga etc is also useful to maintain the fitness in healthy participants.

6. PLOS authors have the option to publish the peer review history of their article (what does this mean?). If published, this will include your full peer review and any attached files.

Reviewer #1: **Yes: **Rober Schleip, Dr. biol. hum.

Reviewer #2: No

Reviewer #3: No

Reviewer #4: No

---

## [Author Response · Author response to Decision Letter 0]

9 May 2022

Response to Review

Manuscript Number: PONE-D-21-20626

Title: "A systematic review of in vivo animal stretching regimens on inflammation and

its relevance to translational yoga research"

Date: 04/29/2022

Editor comments: 

Grant information is now consistently reported in all relevant sections. Specific text for Financial Disclosure now states:

Financial support for the work was provided by grant T32 AT00051 from the National

Center for Complementary and Integrative Health (NCCIH) to DMV, grant

K24AT009282 from NCCIH to PMW, and grant R01DK044128 from the National

Institute of Diabetes and Digestive and Kidney Diseases (NIDDK) to AMZ, grant

K24AT009465 from NCCIH to GYY.

Specific text for Funding Information now states:

This work was supported by T32AT00051 from the National Center for Complementary and Integrative Health (NCCIH: https://nccih.nih.gov/), National Institutes of Health (NIH: http://www.nih.gov/) awarded to DMV. Dr. PW was supported by grant K24AT009282 from NCCIH/NIH. Dr. AMZ was supported by grant R01DK044128 from the National

Institute of Diabetes and Digestive and Kidney Diseases (NIDDK: https://www.niddk.nih.gov/)/NIH. Dr. GYY was supported by grant K24AT009465 from NCCIH/NIH

Data availability statement now indicates the following:

“All relevant data used in this systematic review is included in the manuscript, tables, figures, and in supplementary/supportive information.” Files have been uploaded accordingly. 

Line 687. S1 Table now reads: Electronic search strategy

Reviewer 1:

Q1: We now more clearly address this important distinction beginning on Line 173 in the revised manuscript with the following text:

Active stretching (AS): Refers to a movement applied by an external and/or internal force causing the interaction of the actin and myosin filaments of muscle due to the activation of muscle innervation and targeting the range of motion, i.e., flexibility. 

Passive stretching (PS): Refers to a movement applied by an external force causing the elongation of the myofascial and integumentary systems beyond their resting length and targeting the range of motion, i.e., flexibility.

Q2: We have included this term because in the myofascial system, these nerves will be integral components of muscle spindles and Golgi tendon organs required for proper muscle functioning. Therefore, we consider them part of the myofascial system.

Q3: See changes in manuscript: Lines 294 and 555. 

We appreciate this suggestion and reference to the Hirata et al., 2020 research paper. We now further acknowledge the potential role of age on the inflammatory processes triggered during stretching. 

Reviewer 2:

Q1: Respectfully, at the broader level, we disagree with this comment. The use of active and passive stretching techniques in rodents and different animal tissue/organs has not been systematically synthesized in a review that would allow us to better understand biomechanical cues and their impact on physiological inflammatory processes. More specifically, we hope changes in focus and synthesis in this revised submission enhance the perceived novelty of their contributions. 

Q2: We apologize for this misunderstanding. Indeed, our methods are aligned with the systematic review protocol for animal intervention studies found at the end of de Vries et al., 2015 paper, which is the SYRCLE format. We now include the formal SYRCLE protocol as a supplementary document. This protocol includes our rationale for inclusion/exclusion criteria, search strategy, data extraction, and Risk of Bias (RoB).

We have decided to replace the previous RoB (CAMARADES) for the SYRCLE RoB tool. 

Q3: See changes in manuscript: Line 105 – 120

We have updated our inclusion and exclusion criteria following these valuable recommendations (See SYRCLE protocol).

Q4: We agree that this is a concern. Therefore, to get a sense of the impact of omitting studies that did not specifically include the term stretching, we did a sensitivity analysis including these three additional suggested terms (mobilization, manipulation, manual therapy) between 1900-2020. Two search engines (EMBASE and Web of Science) found 50 and 56 extra articles. Despite this change in number of articles, these additional studies did not change our original search findings or general conclusions. The majority of these new articles (4%) do not explicitly evaluate stretching of the musculoskeletal or integumentary systems. Since yoga’s stretching component is thought to be involved in integumentary and myofascial processes, we consider their relevance to yoga stretching translational research limited. This being said, we now include in our limitation section that we did not include these relevant terms that could change/bias the findings that we report here. 

Similar to the above, our search terms were limited to studies that explicitly used the term “Stretching”. Unfortunately, they may have excluded some DOMS and eccentric muscle contraction studies. We acknowledge that this is a good point and had included and expanded in the discussion the problem with stretching/exercise/and muscle injury models definitions and how they overlap in several ways. Lines: 406 – 433. 

We did not include other animal species because translational studies specifically aiming to inform the impact of stretching in humans have largely rely on rodent models. We acknowledge that other animal stretching studies might exist (e.g., dogs, cats, horses etc.) but they were not meant to explore fundamental mechanisms of stretching. Even though, our search strategy found one pig and three rabbit studies using stretching techniques and evaluating inflammatory outcomes, we have decided to remove them from this systematic review to target only rodents. 

Now, we have removed those terms. 

Lines 663. Now in section ‘Strengths and limitations’ we acknowledge issues regarding librarian involvement and protocol registration. We have updated the periods of search (S1 table). 

Q5: We appreciate these comments related to the rigor of our study.

Regarding a formal protocol, an already included supplementary SYRCLE protocol provides details of our rationale for data extraction (items 31-35; 39-41). 

Outcome measures were also pre-specified including: Inflammatory outcomes measured with different laboratory techniques, such as: mi¬croscopic (e.g., myofiber degeneration), genetic (e.g., gene expression), cell/particle-sorting (e.g., cytokines), enzymatic (e.g., ROS), and macroscopic techniques (e.g., muscle or connective tissue ultrasound).

We were interested in extracting information regarding the following stretching parameters: intensity, duration, and frequency. 

We classified studies as short- or long-term studies for the duration parameter. 

We classified studies as either single or multiple bouts of stretching for the frequency parameter. 

For results, we only focus on narrative results. Due to the heterogeneity of these studies (e.g., type of intervention, stretching parameters, inflammatory outcomes), we did not extract continuous/dichotomous numerical data and could not employ meta-analytic techniques.

Lines 128 and 136. As noted in the methods section of the manuscript, two reviewers (DMV and WG) extracted all data. Discrepancies were resolved by discussion, and if an agreement was not reached, these were further discussed in a meeting with two additional independent reviewers (PW and GY).

Q6: We have decided to replace the previous RoB (CAMARADES) for the SYRCLE RoB tool

Q7: See changes in manuscript: Lines 1-3. 

We appreciate this comment, but as is apparent in our narrative, a major impetus for this work is to synthesize and link work done on tissue stretching as it relates to future translational research in yoga like practices. We have modified the title of this paper to: 

A systematic review of in vivo stretching regimens on inflammation and its relevance to translational yoga research

Reviewer 3:

Q1: We appreciate this helpful perspective/concern. We now more clearly address the relevance of specific pre-clinical (in vivo) studies to help narrowing the current knowledge gap that exist between the isolated component of yoga (i.e., yogic stretching) and its interaction with systemic inflammatory, regenerative, and remodeling processes. 

In the manuscript (discussion section), we now more comprehensively discuss research on the effect of yoga on inflammation in humans, as well as challenges in study design. 

Lines: 363 - 404 

Q2: We have decided to replace the previous RoB (CAMARADES) for the SYRCLE RoB tool.

Q3: See changes in manuscript: Line 173. 

Active stretching (AS): Refers to a movement applied by an external and/or internal force causing the interaction of the actin and myosin filaments of muscle due to the activation of muscle innervation and targeting the range of motion, i.e., flexibility. 

Passive stretching (PS): Refers to a movement applied by an external force causing the elongation of the myofascial and integumentary systems beyond their resting length and targeting the range of motion, i.e., flexibility.

Q4: See changes in manuscript: Lines 250 - 268. 

We have harmonized tables 3 and 4 following this reviewer’s recommendations. 

Q5: The studies summarized here showed great heterogeneity within each key protocol feature discussed in the manuscript. For example, in passive vs. active stretching protocols, stretching parameters, injurious vs. therapeutic stretching paradigms. We have also avoided using a quasi-quantitative numerical summary(5/6), as suggested by the reviewer in the revised manuscript. 

Instead of a meta-analysis, we aimed to narratively summarize different inflammatory outcomes used by different stretching techniques of the musculoskeletal and integumentary systems of in vivo rodent models.

Q:6 As noted above, in our discussion we have significantly expanded on the state of the knowledge of yoga clinical research and inflammation through summaries provided by recent systematic reviews. We have subtitled this section: “Current knowledge gaps between the isolated physical component of yoga (i.e., yogic stretching) and its interaction with systemic inflammatory, regenerative and remodeling processes.” We consider that a more comprehensive summary of this clinical literature is outside of the scope of this systematic review. 

In this section, we are careful to point out that in the majority of these studies the effects of mechanical forces cannot be isolated from other potentially therapeutic components of yoga (e.g., focus attention, meditation, breathing exercise). Because of this, we respectfully request to leave some mention of future experiments in animals. Currently, through animal studies, with carefully chosen models and following animal welfare policies (e.g., 3Rs), we will be able to isolate these specific processes. Towards that end, we wish to convey that in addition to future clinical studies, future research with animals will remain critical, and together will support a bidirectional translational model that we believe will move the research field forward. 

Lines: 363 - 404 

Q7: See changes in manuscript:

Line 117: Studies before 1900 were excluded because the laboratory techniques to evaluate inflammatory outcomes were not yet fully developed.

Lines 73: We have now included references that supports this statement. 

Line 87: we have removed this term.

Line 92: We have changed these terms as suggested. Also, we included our formal SYRCLE protocol as a supplementary document. In this protocol, it can be found our rationale for inclusion/exclusion criteria, search strategy, data extraction and RoB.

Line 112: As noted above, we did not include other animal species because translational studies specifically aiming to inform the impact of stretching in humans have largely rely on rodent models. We acknowledge that other animal stretching studies might exist (e.g., dogs, cats, horses etc.) 

Lines 105 – 120: We have rephrased our inclusion and exclusion criteria section accordingly.

Line 128 – 136: Two reviewers (DMV and WG) extracted all data. Discrepancies were resolved by discussion, and if an agreement was not reached, these were further discussed in a meeting with two additional independent reviewers (PW and GY). This information is included in the SYRCLE protocol, which is included as a supplementary document. 

Line 138: We have decided to replace the previous RoB (CAMARADES) for the SYRCLE RoB tool. 

L138: we have changed our RoB to SYRCLE.

L352: This has been corrected.

We agree with the suggestion; figure 4 has been deleted. 

Figure 1 has been updated accordingly.

Reviewer 4:

Q1: First, we fully agreed with this concern, and one of the main conclusions of this review is that novel paradigms that evoke or include active stretching in rodents and other animals will build stronger bridges between animal and clinical work (e.g., self-stretching animal paradigms)

Second, that being said, at the most fundamental level, we argue that stretching of the extracellular matrix/connective tissue/fascial system (i.e., collagen and elastin fibers and resident cells, such as myocytes, fibrocytes, tenocytes, chondrocytes, osteocytes, endothelium cells, etc., and their blast members) occurs either during active or passive stretching. 

Third, we tried to clarify our use of the terminology from Sports medicine literature for stretching in table 2. Line 173 and included more nuances in our discussion around active versus passive stretching. Line 406.

Lines 406 – 484: These lines address partly this concern. 

Q2: We fully appreciate this concern. Yoga is a complex intervention that include many other potentially therapeutic components above and beyond stretching. It is precisely for these reasons that we looked the rodent studies to better understand the potential isolated effects of mechanical physical cues (i.e., stretching) on inflammation. With this understanding, we hope to suggest more complex animal paradigms controlling for other confounders such as the stress during animal manipulation or during the anesthetic procedure. Similarly, we think this information can inform bidirectionally the design of clinical studies controlling for some non-mechanical component of yoga (e.g., focus attention). 

This distinction is now better articulated in our discussion. Lines: 363 – 404

Q3: We acknowledge that the integumentary and musculoskeletal system are typically consider two “different body systems”. However, the two systems are interconnected inside through a mesh of connective tissue, which permeates each body part and creates the necessary organic scaffold for sustaining more specialized cells (e.g., myocytes, satellite cells, immune cells, keratinocytes, etc.), and therefore creating a unity. 

Since during yogic stretching and passive/active stretching in animals, the two main “systems” being stretched are the integumentary and musculoskeletal system, we decided to include studies targeting these systems. 

During stretching (both in animals and humans), it is likely that mechanical cues are applied in a continuum between the epidermis, dermis, subcutaneous, fascia, muscle and bone. 

Q4: See changes in manuscript lines: 451 – 453; 473 – 484; 656 – 661. 

These lines, in part, address this sharp observation. We understand that a main caveat of stretching animal studies is the contribution of stress either during active stretching (human manipulation) or passive stretching (anesthetic procedure) in the subsequent inflammatory process. 

The stress can’t not be ignored and can confound the effects of stretching on inflammation. We discussed in lines: 471 – 477 the need of active self-stretching paradigms to eliminate the human manipulation interaction. Such paradigm would successfully keep only the physical stress of stretching.

Q5: This word has been delated. Line 87

---

## [Editor Report · Decision Letter 1]

19 May 2022

A systematic review of in vivo stretching regimens on inflammation and its relevance to translational yoga research

PONE-D-21-20626R1

Dear Dr. Munoz Vergara,

We’re pleased to inform you that your manuscript has been judged scientifically suitable for publication and will be formally accepted for publication once it meets all outstanding technical requirements.

Kind regards,

Robert Schleip

Guest Editor

PLOS ONE

Additional Editor Comments (optional):

Dear authors

Thank you very much for your detailed response, in which you successfully addressed all points addressed in the review process.
---

## [Editor Report · Acceptance letter]

23 May 2022

PONE-D-21-20626R1 

A systematic review of *in vivo* stretching regimens on inflammation and its relevance to translational yoga research 

Dear Dr. Muñoz-Vergara:

I'm pleased to inform you that your manuscript has been deemed suitable for publication in PLOS ONE. Congratulations! Your manuscript is now with our production department. 

Kind regards, 

on behalf of

Dr. Robert Schleip 

Guest Editor

PLOS ONE